# High-speed optical imaging with sCMOS pixel reassignment

**Biagio Mandracchia** [1,2,5], **Corey Zheng** [1,5], **Suraj Rajendran** [1], **Wenhao Liu** [1], **Parvin Forghani** [3], **Chunhui Xu** [3,4] & **Shu Jia** [1,4] ✉

Fluorescence microscopy has undergone rapid advancements, offering unprecedented visualization of biological events and shedding light on the intricate mechanisms governing living organisms. However, the exploration of rapid biological dynamics still poses a significant challenge due to the limitations of current digital camera architectures and the inherent compromise between imaging speed and other capabilities. Here, we introduce sHAPR, a high-speed acquisition technique that leverages the operating principles of sCMOS cameras to capture fast cellular and subcellular processes. sHAPR harnesses custom fiber optics to convert microscopy images into one-dimensional recordings, enabling acquisition at the maximum camera readout rate, typically between 25 and 250 kHz. We have demonstrated the utility of sHAPR with a variety of phantom and dynamic systems, including high-throughput flow cytometry, cardiomyocyte contraction, and neuronal calcium waves, using a standard epi-fluorescence microscope. sHAPR is highly adaptable and can be integrated into existing microscopy systems without requiring extensive platform modifications. This method pushes the boundaries of current fluorescence imaging capabilities, opening up new avenues for investigating high-speed biological phenomena.

Complementary metal-oxide-semiconductor (CMOS) technology has revolutionized optical imaging by facilitating the development of a new generation of detectors that offer higher framerates compared to their predecessors based on charge-coupled devices[1]. CMOS sensors are generally compact and cost-effective, allowing for the integration of more analog-to-digital converters on the chip and faster pixel readouts via rolling-shutter capture. In particular, recent advances in scientific-grade CMOS (sCMOS) technology present a remarkable signal-to-noise ratio (SNR), approaching the imaging performance of true low-light detectors with a high quantum efficiency (>90%) and low readout noise (<2e⁻)[2]. In addition, modern sCMOS cameras feature large imaging arrays (>4 megapixels), 16-bit pixel depth, and high full-sensor readout speeds over 100 Hz[3]. Due to these capabilities, sCMOS cameras have been widely adopted for fluorescence microscopy[4,5], paving the way for a diverse array of biological discoveries with unprecedented details such as functional imaging, signal transduction, subcellular dynamics, and membrane trafficking[6-13].

Despite these advances, capturing rapid biological events at the cellular and subcellular levels beneath the sub-millisecond regime remains a considerable challenge for fluorescence microscopy. Current strategies may necessitate a compromise of imaging abilities, such as speed, resolution, and field of view (FOV)[14], or alternatively, the use of unconventional frameworks[15-21], which may, in turn, add instrumental complexity, limit accessibility to commonplace platforms (e.g., epifluorescence microscopes), or present inherent constraints due to mechanical inertia, anisotropic resolution, and image distortion[18-20,22-24].

[1]Wallace H. Coulter Department of Biomedical Engineering, Georgia Institute of Technology and Emory University, Atlanta, GA, USA. [2]E.T.S.I. Tele-comunicación, Universidad de Valladolid, Valladolid, Spain. [3]Department of Pediatrics, School of Medicine, Emory University, Atlanta, GA, USA. [4]Parker H. Petit Institute for Bioengineering and Bioscience, Georgia Institute of Technology, Atlanta, GA, USA. [5]These authors contributed equally: Biagio Mandracchia, Corey Zheng. ✉e-mail: shu.jia@gatech.edu

Indeed, sCMOS cameras face comparable constraints akin to their counterparts in the context of fast imaging. sCMOS sensors permit enhancing the time resolution by selectively reducing the number of active pixel rows. Typically, with their high line readout rates surpassing 100 kHz, sCMOS cameras can attain around 1 kHz[25,26] when tailored to a single-cell FOV (e.g., ~100 × 100 pixels, or tens of micrometers, Fig. 1a). However, owing to the column-parallel analog-digital converter architecture typical of CMOS sensors[27,28], framerates are dependent on the number of pixel rows rather than the total number of pixels. As a result, the pursuit of higher framerates with this configuration may lead to an undesirably narrow FOV, obstructing the visualization of wide-field, isotropic biological features, even at low magnification. Alternatively, the full sCMOS sensor area has been harnessed to increase the imaging throughput[22,29,30] without increasing framerate, but the benefits of the augmented throughput are largely confined to specific applications such as microfluidics[31]. Further attempts attained a higher temporal resolution for fluorescence imaging using space-to-time multiplexing, based on a reimaging optical cavity[32] or a digital micromirror device[33]. However, these solutions are restricted to operating in burst mode, i.e., providing only a limited series of consecutive high-speed frames, which may undermine their wider applicability for longitudinal observation.

To overcome these limitations, we present sCMOS high-speed acquisition with pixel reassignment (sHAPR). We introduce an integrated optical and computation approach (Fig. 1a) that maximizes the sensor framerate for small and isotropic FOVs through careful consideration of CMOS detector architectures. In sHAPR, a custom fiber bundle is employed to spatially rearrange the light received from the imaging system, converting it into a monodimensional image. Subsequently, this image is projected onto a linear region of the detector operating in a reduced readout mode, enabling high-speed recording at the microsecond scale (from ~25 to 250 kHz, depending on the sCMOS sensors). Then, a computationally efficient algorithm reverts the acquired image back to its original undistorted form. Notably, sHAPR is independent of the sample illumination method employed and thereby compatible with virtually any existing microscopy

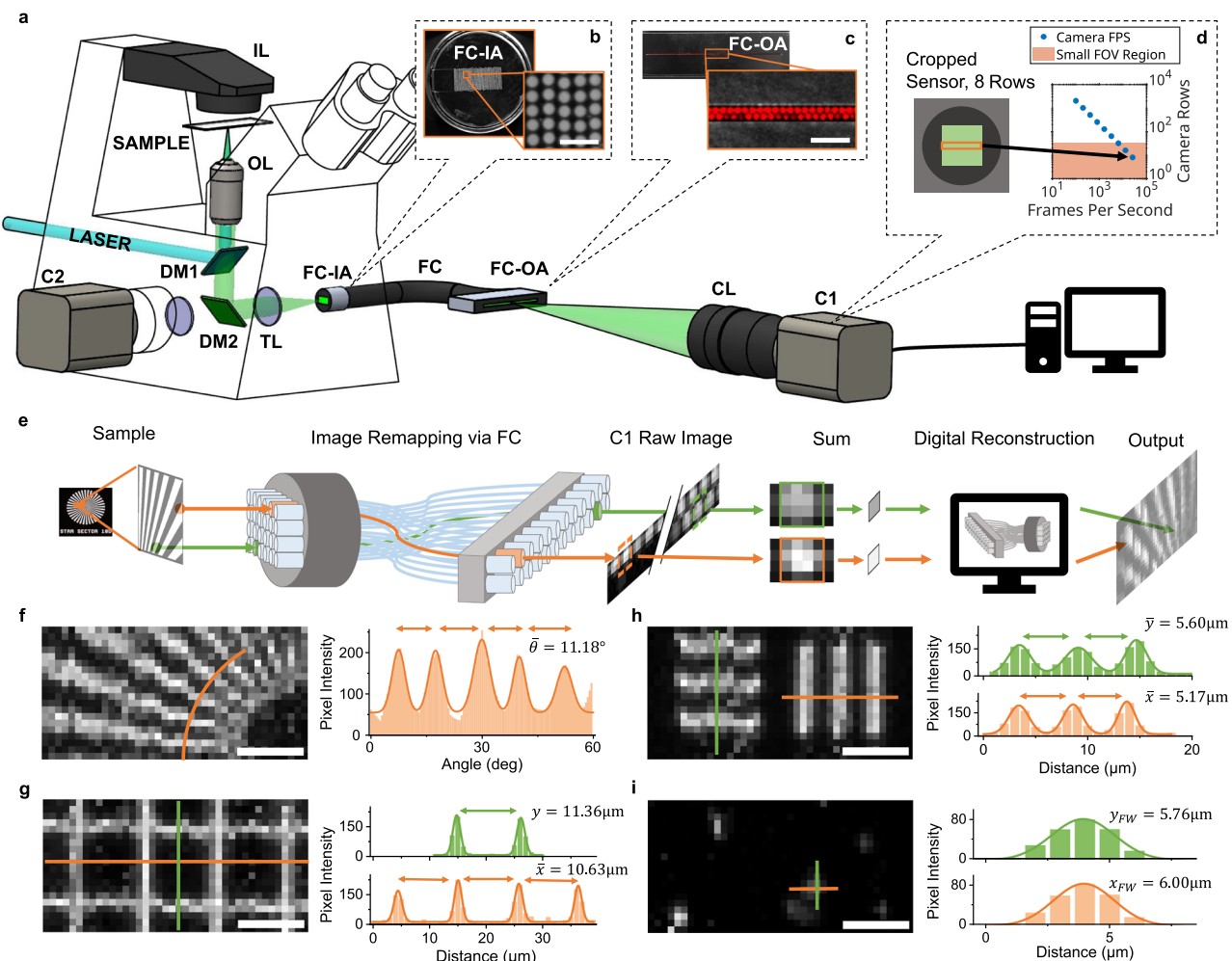

**Fig. 1 | sCMOS high-speed acquisition with pixel reassignment (sHAPR).**
**a** Schematic of the experimental setup. DM dichroic mirror, TL tube lens, OL objective lens, IL illumination lamp, FC fiber coupler, FC-IA fiber coupler input array, FC-OA fiber coupler output array, C camera, CL camera demagnification lens. **b** Close-up image of the physical FC-IA. **c** Close-up image of the physical FC-OA. **d** Depiction of the relationship between subarray size (in rows) and camera FPS. **e** Imaging flowchart. From left to right: the Siemens Star is taken as an example input sample; Diagram showing the mapping of fibers of the FC-IA to FC-OA; Cropped sample of the raw image of FC-OA captured by C1; Demonstration of final image reconstruction process involving summing fiber signals and assigning final image position based on the pre-calibrated spatial relationship between the FC-IA and FC-OA. **f–i** Calibration images of various optical targets. **f** Siemens Star with 36 bars of 10°; measured bar angular separation = 11.18°. **g** Thorlabs optical target 10 μm pitch grid with 1.5 μm line width; measured peak-to-peak x separation = 10.63 μm, y separation = 11.36 μm, mean x FWHM = 1.60 μm, mean y FWHM = 1.64 μm. **h** USAF 1951 target group 7 element 5 with 2.46 μm line width; measured peak-to-peak x separation = 5.17 μm, peak-to-peak y separation = 5.60 μm. **i** 4-μm green TetraSpeck fluorescent bead; x FW = 6.00 μm, y FW = 5.76 μm. Colors correspond to profile lines. Scale bars: 400 μm (**b**), 800 μm (**c**), 10 μm (**f–i**). Source data are provided as a Source Data file.

systems, allowing for enhanced imaging capabilities without the need for extensive platform modifications. To show its potential in a wide range of applications where high-speed recording is critical, we implemented the method on a commonly-used epifluorescence microscope and used it to perform high-throughput flow cytometry, imaging of contraction in cultured cardiomyocytes, and tracking of neuronal calcium waves. To the best of our knowledge, this is the fastest direct imaging of neuronal action-potential-driven calcium waves reported using fluorescence microscopy.

## Results

### Principle and framework of sHAPR

Conventionally, sCMOS cameras can attain ultrahigh framerates by selecting narrow regions of interest (ROIs) where the height is confined to a limited span of pixels. In this work, sHAPR exploits optical pixel reassignment to leverage the inherent high-speed row-readout capabilities of the CMOS sensors for wide-field acquisition. In particular, optical pixel reassignment is achieved using a customized fiber bundle, which is configured in a 2D layout for capturing the image and subsequently reorganized into a 1D array, enabling precise projection onto a selected number of active pixel lines on the sCMOS sensor (Fig. 1a).

As depicted in Fig. 1a and Fig. S1, the sHAPR system is constructed on top of an epifluorescence microscope, which is equipped with 100× 1.45NA and 40× 1.3NA objective lenses. The samples are illuminated with either a laser line (488 nm) or the built-in broadband white lamp of the microscope. A custom fiber coupler (FC), composed of 800 optical fibers, is employed to effectuate pixel reassignment (detailed in Supplementary Section 2.1). On the input array of the FC (FC-IA), these optical fibers are organized into a 20 × 40 rectangular matrix, positioned at the intermediate image plane of the microscope (Fig. 1b). The intercepted image is then relayed through the FC to the output array (FC-OA), where the fibers are reconfigured into a slim 2 × 400 strip (Fig. 1c). The dimensionally remapped image emerging from FC-OA is relayed through a coupling lens (CL) and captured by an sCMOS camera, which operates in a reduced readout mode at its narrowest allowable ROI of 8 × 2048 pixels, thus facilitating the acquisition of up to 25,600 frames per second (Fig. 1d). To make full use of the available rows in the ROI, the magnification of the CL is finely tuned such that each fiber end at the FC-OA is projected to a 4 × 4-pixel area on the camera sensor, easing alignment, and increasing dynamic range (Fig. 1e).

Then, the image captured by the camera undergoes post-processing to retrieve the original signal from the microscope, as outlined in the flowchart of Fig. 1e. In practice, pixel intensity is binned within each 4 × 4-pixel region that corresponds to an individual fiber end, and the resultant 1D signal is assigned to the respective position in the original 2D coordinate system. This allocation is executed based on a pre-calibrated map that establishes the correlation between fiber positions on FC-OA and FC-IA (Supplementary Section 2.2 and Fig. S2), thereby synthesizing the original 2D image with computational efficiency.

To test the system, we initially employed the 100× objective lens and 1.5× microscope magnification dial and captured images of various static optical targets with a 1-ms exposure time, which yields a pixel size of 1 μm and a 20 μm × 40 μm FOV. As shown, the system effectively reconstructed brightfield images of commonly-used test targets, such as a Siemens star (Fig. 1f), a 10-μm grid (Fig. 1g), and the group 7 element 5 of a USAF 1951 target (Fig. 1h), as well as fluorescent images of 4-μm TetraSpeck beads (Fig. 1i). We measured the peak-to-peak distance, angular separation, and the full width (FW) of the profiles using Gaussian fitting, quantitatively verifying their consistency with the expected target dimensions. In these given experiments, for instance, the recovered 10-μm grid and the USAF target provided pitch estimates of 11.0 and 5.39 μm, matching the nominal values of 10 and 4.92 μm, respectively, within an error margin of 1 pixel. We also

performed high-speed imaging of moving targets at 25.6 kHz, including bubbles and sand grains, within a microfluidic platform (Supplementary Movie 1).

As demonstrated, the incorporation of the fiber-optic bundle facilitates acquisition at ultrahigh speeds while capturing an isotropic FOV across lateral dimensions, minimizing discrepancies well below the margin of measurement error. This enhancement applies seamlessly to both brightfield and fluorescent targets, enabling sHAPR to surpass the limitations of traditional sensor cropping in standard sCMOS camera-based systems. For instance, compared to selecting a sensor with a 20 × 40 pixel ROI, sHAPR showcases a framerate increase between 2.5 and 10 times, depending on the camera model. These performance gains become even more substantial with larger fiber bundles and more uniformly isotropic FOVs (Supplementary Table S2, Supplementary Section 2.4, 2.5).

### High-throughput imaging flow cytometry

Imaging flow cytometry (IFC) combines flow cytometry and optical microscopy to enable high-throughput multiparametric single-cell analysis with rich spatial details, high sensitivity, and molecular specificity[23,31,34,35]. IFC technologies have been applied in various basic and translational fields, including cell biology[36], immunology[37,38], microbiology[39], hematology[40], and cancer research[41,42]. However, a notable challenge arises when employing common camera-based microscopes, as they often lack the required recording speed for high-throughput screening. Indeed, to achieve a high throughput (e.g., over 10,000 cells/s), it is generally necessary to adopt either instrumentally complex solutions such as time-encoded amplified microscopy[43], radiofrequency-tagged emission[19,31], and virtual-freezing fluorescence imaging[44], or take advantage of specific microfluidic architectures as in the case of massively parallel microfluidic channels[29,30]. On the contrary, sHAPR offers a straightforward high-speed cytometric imaging strategy with wide compatibility across both imaging techniques and microfluidic platforms.

To validate sHAPR for high-throughput IFC, we imaged nucleus-labeled HeLa cells flowing through a microfluidic channel. To reduce motion blur, we implemented stroboscopic illumination with a duration of 5 μs (Supplementary Section 3.1)[30]. This illumination was achieved using an AOTF controlled by square pulses from an external function generator (Fig. 2a). The experiments were then carried out with a range of flow speeds and objective magnifications, as illustrated in Fig. 2b–e. We achieved a high flow speed of ~1 m/s without noticeable image distortion or motion blur (Fig. 2e inset, Fig. 2f, and Supplementary Movie 2). This results in an analytical throughput of 25,000 cells/s, derived from extrapolations by assuming that the cells are spaced on an average of 40 μm (>2× the cell size[45]) with our current setup. Moreover, through finer control of the camera settings and stroboscopic illumination, our current system can be further optimized to approach the theoretical maximum flow speed permitted by the FOV and frame rate, achieving far greater throughput of 38,400 cells/s and beyond depending on magnification[45] (Fig. 2g, Supplementary Section 3.3, 3.4). sHAPR is well-positioned to facilitate its adoption through the approachable detector-based modification that can support a mixture of analytical techniques for cellular diagnostics[31].

### Tracking calcium transients in cardiomyocytes

Human induced pluripotent stem cell-derived cardiomyocytes (hiPSC-CMs) have emerged as a vital source for in vitro modeling of drug-induced cardiotoxicity, genetic cardiovascular disorders, drug screening, and in vivo cardiac regeneration research[46]. The 3D microtissues of hiPSC-CMs offer a promising platform to study cardiac physiology, regeneration, disease, and pharmacology[46]. Meanwhile, fluorescence microscopy has become the technique of choice for the investigation of the intricate phenotypic and functional characteristics

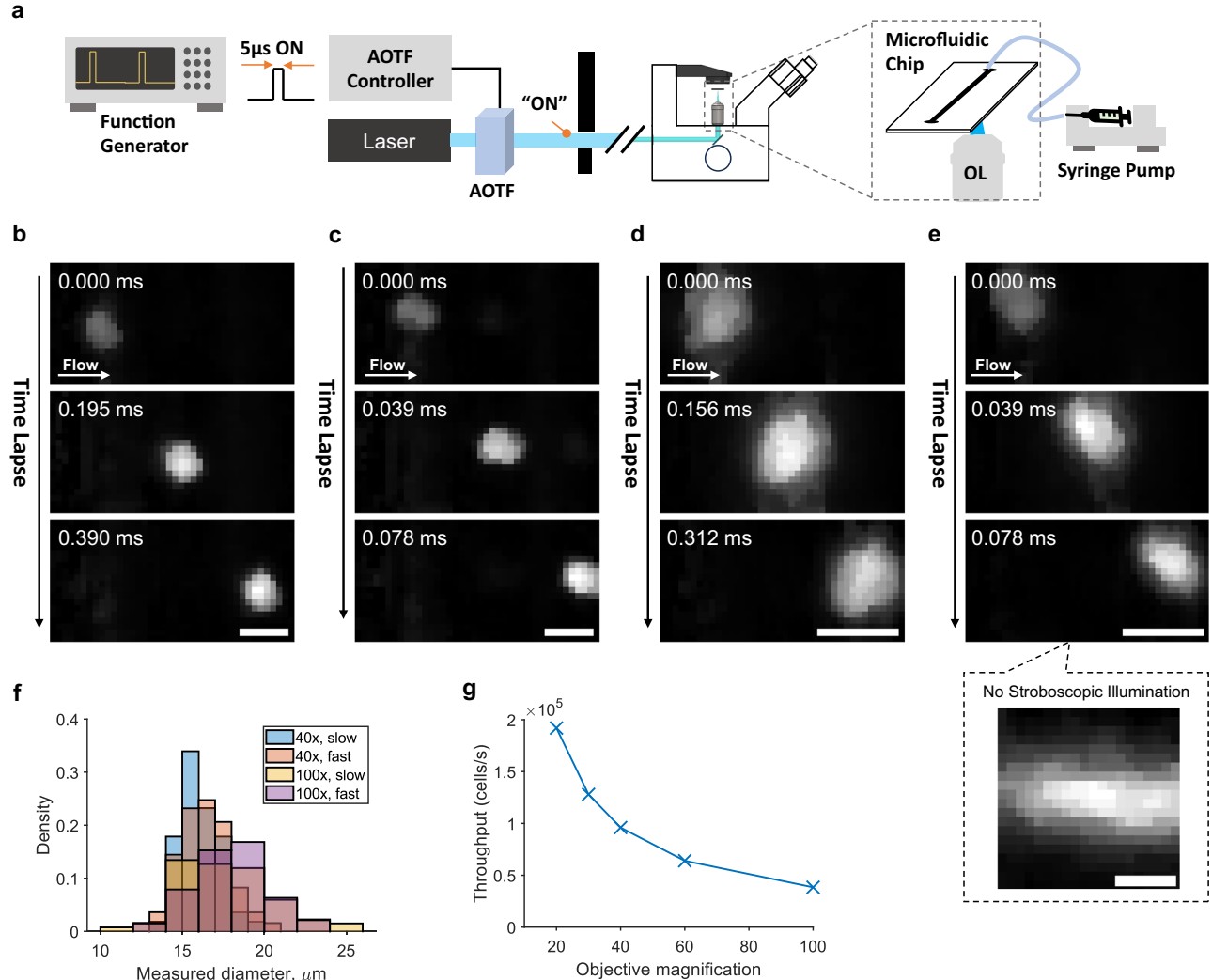

**Fig. 2 | Imaging flow cytometry with sHAPR. a** Schematic of microfluidic imaging setup using sHAPR, containing a stroboscopic illumination setup using AOTF to shutter excitation laser for high-speed recording. **b–e** HeLa cells stained with Syto-16 flowing captured at varying flow speeds and magnifications (0.16 m/s, 40×; 0.90 m/s, 40×; 0.12 m/s, 100×; 0.50 m/s, 100×, respectively) at the framerate of 25.6 kHz with a 5-μs stroboscopic illumination duration. **e**, inset: cells captured under the same setting without stroboscopic illumination, displaying the motion blur. **f** Diameter distribution of the nucleus for each experimental group (each group size >50 cells). **g** Analytical cell throughput at an assumed cell spacing of 40 μm. Scale bars: 20 μm (**b–e**), 10 μm (**e** inset). Source data are provided as a Source Data file.

of hiPSC-CMs[47–49]. However, current imaging methods of cellular signaling propagation, such as confocal and light-sheet microscopy, are still limited by either the pixel dwell time (≥1 μs)[50–53] or the camera speeds[10,54–56].

Here, we demonstrated high-speed sHAPR imaging of calcium signaling of hiPSC-CMs. Experimentally, we arranged hiPSC-CMs on a monolayer, labeled them with calcium indicators, and imaged the calcium transients at different framerates. First, we recorded spontaneous excitation-contraction transients at a sampling rate of 1 kHz, revealing the spatial propagation of the calcium transient across the cellular space (Fig. 3a–c and Supplementary Movie 3). Meanwhile, the fluctuating signals of the calcium transient stemming from each cell can be extracted and traced (Fig. 3d), and the key parameters, such as the rise time to peak ($t_r$), period ($t_p$), and maximum fluorescence ($F_{max}$), were calculated using electrophysiology software[57] further described in the Methods section. Our results yielded the mean values of $t_p = 2.71 \pm 0.05$ s, $F_{max} = 1.80 \pm 0.41$, and $t_r = 119.5 \pm 27.2$ ms ($n = 3$ cells), aligning well with previously reported studies[58,59] and thereby demonstrating the consistency and reliability of sHAPR for studying excitation-contraction transients.

In particular, we were able to record regions that exhibited distinct calcium sparks at high acquisition rates of up to 25.6 kHz (e.g., at least three active regions in Fig. 3e and Supplementary Movie 4). As seen, the calcium signals in the region near the cell membrane (e.g., Region 1 in Fig. 3e) displayed a large subcellular wave-like propagation of synchronous calcium spark activation, while the regions within the cell body (e.g., Regions 2 and 3 in Fig. 3e) were more isolated, showing comparatively lower intensity and reduced spatial propagation (Fig. 3f, g). The corresponding patterns displayed in these regions ranged from relatively rhythmic waves, occurring at an interval of approximately 300 ms from the cell periphery, to more irregular activities (Fig. 3h). Compared to previously documented mammalian calcium spark behaviors using confocal microscopy (recorded at less than 500 Hz), our system is capable of observing similar elementary calcium spark phenomena, including spark repetition (Fig. 3g, h) and leading of full-cell waves (Supplementary Movie 5)[60–62]. Lastly, we demonstrated calcium sparks from hiPSC-CMs captured at differing acquisition rates (Fig. 3i), verifying the high-speed ability of sHAPR to preserve the consistency across multiple time scales and delineate the exponential-like rise and decay

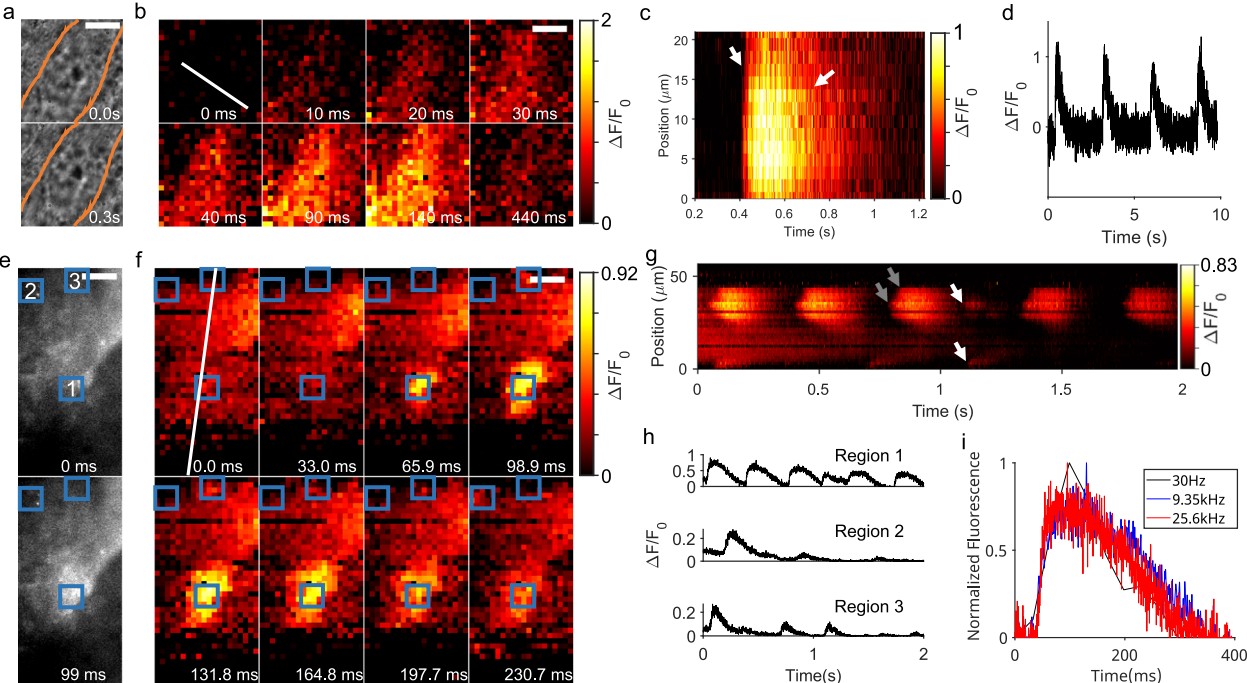

**Fig. 3 | Imaging cardiomyocyte activity with sHAPR. a** Brightfield images of cardiomyocytes before and during spontaneous contraction. The region of interest is outlined in orange. **b** Time-lapse images of fluorescent activity during contraction captured at 1000 Hz. **c** Kymograph of single contraction along profile indicated by the dotted line in (**b**), showing a spatial propagation across the profile during the fluorescence rise time, along with a spatially-varying decay in fluorescence. **d** Representative fluorescent activity across four spontaneous contractions measured from a single pixel. **e** Fluorescent wide-field images taken from the low-speed camera C2 at 30 Hz before and during a calcium spark. Three regions where calcium sparks occur are indicated using blue boxes (1–3). **f** Time-lapse images of

multiple calcium sparks within the same cardiomyocyte captured at 9.35 kHz. The three corresponding regions matching those in (**e**) are outlined in blue boxes and sparking activity in regions 1 and 3 can be observed in the time-lapse. **g** Kymograph of calcium spark activities along the profile indicated by the white dotted line in (**f**). Arrows indicate two regions of sparks, where spatial propagation (gray) and temporal irregularity (white) were observable. **h** Comparison of fluorescent activities captured at 9.35 kHz in each region indicated in (**f**). **i** Comparison of the fluorescent activity for a single calcium spark occurring in region 1 at multiple acquisition speeds. Scale bars: 10 μm. Source data are provided as a Source Data file.

processes, in comparison with the epifluorescent data taken at a lower speed of 30 Hz.

## Imaging action-potential-driven calcium waves in neurons

Neurons constitute extraordinarily dynamic cell systems that support intricate patterns of communication. In particular, the neuronal action potential (AP), or spike, serves as the primary means of signal propagation across neuronal networks. They exhibit significant variation in their properties, including frequency[63], directionality[64], and origin[65], leading to a rich field of study aimed at elucidating the mechanics behind information propagation through biological systems of varying spatial scales. Fluorescent imaging has become a key technology for this field, providing a non-invasive alternative to traditional electrode probing for examining neuronal dynamics[66,67], including demonstrations in neuronal network communication[68,69], functional imaging[70,71], and subcellular phenomena[72]. Importantly, improved temporal resolution within fluorescent imaging has unveiled new details about AP characteristics[68,73–75]. While neuron imaging beyond 10 kHz has previously relied on sub-frame interpolation[73–75], sHAPR can achieve similar framerates with direct imaging.

Here, we demonstrated the use of sHAPR for capturing high-speed AP-associated calcium dynamics in neurons. Fluorescent calcium imaging is a common approach to capturing the dynamics of neuron signaling, as the AP of an excited neuron triggers the influx of calcium ions through various channels[67]. Specifically, cultured rat hippocampal neurons were labeled with Fluo-4 calcium dye and subsequently stimulated using a lab-built electronic probe[76] (detailed in Supplementary Section 5.1), with each stimulation producing an

AP-evoked calcium transient within the neurons (Fig. 1a and Supplementary Movie 6). We performed imaging synchronously with stimulation pulses using the high-speed sHAPR system at a rate of 25.6 kHz. To improve the SNR, at least ten AP sequences were averaged for each neuron (Supplementary Movie 7). Within 10 ms, fluorescent signals rise to the near-maximum (Fig. 4b), emphasizing the rapid dynamic details necessitating high acquisition rates. To probe the propagation characteristics, we curve-fitted the rising portion of the fluorescent time trace for each pixel (Fig. 4b). From these fittings, we extracted the time at which each pixel reached 50% of its peak value and generated isochronal maps for each neuron (Fig. 4c–e and Supplementary Section 5.2). From the resultant maps, we can distinguish the direction of propagation of the calcium signal across the neurons. Notably, the calcium signal appeared to propagate outward from the junction between a dendrite and a soma (Fig. 4c and Supplementary Movie 8), while other cells exhibited propagation from peripheral dendrites towards the center of the soma (Fig. 4d and Supplementary Movie 9). We also observed the propagation of the signals from the edge to the center of the soma (Fig. 4e and Supplementary Movie 10). Such a breadth of signaling variety results from an intricate combination of underlying variables[66], including the field stimulation that can excite various parts of the neuron[76], natural biological complexity involving the relative concentration and location of membrane voltage-gated calcium channels[77,78], and calcium kinetics[79]. To the best of our knowledge, this study presents the demonstration of the direct visualization of AP-driven calcium wave propagation matching framerates previously achieved only using sub-frame interpolation[73–75].

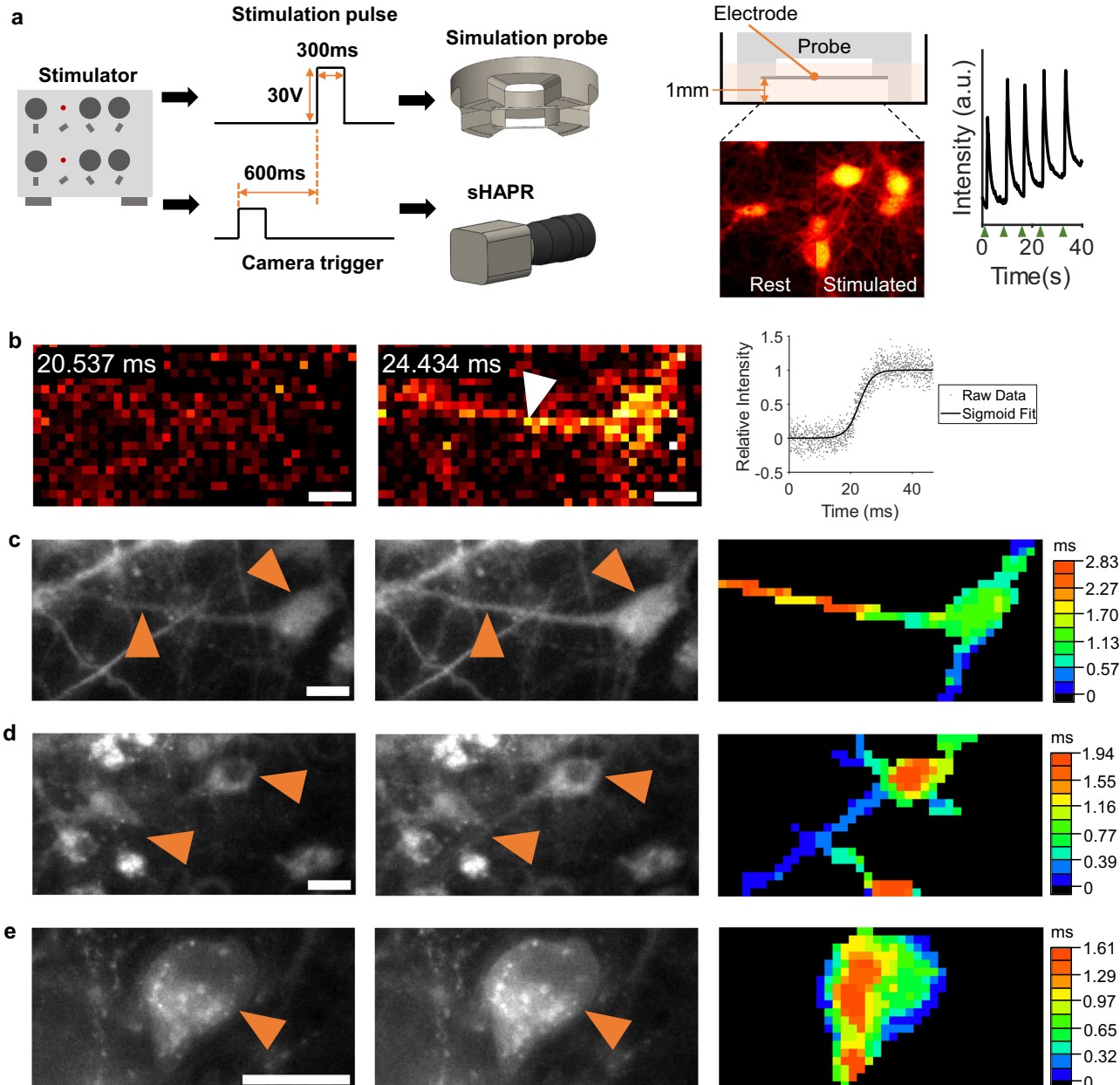

**Fig. 4 | Imaging neuronal calcium waves with sHAPR. a** Neuron stimulation setup for sHAPR. Left diagram: probe stimulation pulse and acquisition pulse timing diagram. Right diagram: Stimulation probe position in neuronal culture dish and pre- and post-stimulated neuron fluorescent images. Right plot: average neuron fluorescent activity in response to manual electrical stimulation at times marked by green arrows (a.u.: arbitrary units). **b** Unprocessed video stack containing >10 excitations of the same Fluo-4 stained neuron. Right plot: Normalized signal from a single pixel indicated by a white arrow fitted to a sigmoidal curve. **c**–**e** Three sets of neuron excitations were captured with sHAPR. Left: neurons before stimulation captured by the low-framerate camera. Middle: neurons during stimulation captured by the low-framerate camera. Right: isochronal maps generated using high-speed video fit to sigmoidal excitation curves. Isochronal map timestamps mark when a pixel has reached 50% of its peak value. Maps are masked to the neuron body using high-resolution images captured by the low-framerate camera. Maps reveal a variety of calcium fluorescent signal propagations, presenting movement along processes (**c**, **d**) and into the cell body (**d**, **e**). Scale bars: 20 µm. Source data are provided as a Source Data file.

We expect the applications of our technology to extend beyond calcium imaging. High-speed voltage imaging has enabled the direct capture of high-speed electrical signals using indicators with microsecond response times[80,81], opening the door to signaling phenomena that cannot be captured by calcium influx indicators[66,82,83]. As such, it is increasingly applied toward neural imaging. A clear challenge towards enabling this high-speed voltage imaging is the lower signal level of voltage indicators[82–84] and low exposure times associated with rapid framerates, limiting current imaging studies to approximately 10kHz[74] rates while also requiring trial averaging to increase SNR[74,75]. As

demands for capturing dynamic and evolving neuronal activity (e.g., rapid oscillations and animal activities) increase, along with requirements for >20 kHz acquisition speeds to fully sample the highest frequency component of APs[85], sHAPR is well-positioned to quickly adapt to and capitalize on next-generation low-light imagers. New CMOS cameras with extraordinarily low readout noise, high QE, and even integrated image intensifiers have been developed and can be swapped into the sHAPR setup to boost the signal. Whilst current system losses in the fiber bundle and an outdated camera model limit the signal collection ability of our current implementation, future

improvements are easily achievable (Supplementary Section 2.5), and sHAPR can pave a promising avenue toward further explorations of complex interactions, ranging from kinetics to biophysics and intracellular communications.

## Discussion

In recent years, 2D scientific detectors have achieved single-photon sensitivity, but their recording speed remains constrained compared to that of industry-grade cameras. However, all the attempts to bypass this shortcoming presented several limitations, such as the ability to provide only brief bursts of frames, short exposure dwell-times, or the incompatibility with a wide range of imaging systems[86].

In this study, we utilized image reformatting with flexible fiber optic pixel reassignment to take advantage of the rapid line readout capability inherent in sCMOS sensors. Despite their wide-ranging applications in imaging and sensing, such as endoscopy and spectroscopy, among others[87–100], fiber optic relays have not previously been harnessed to exploit this feature. Thus, sHAPR presents users with the opportunity to enhance imaging speed while having the flexibility to rearrange pixels to achieve the most suitable FOV for their application. Compared to conventional camera cropping, sHAPR offers a significant improvement when very high speeds (>20 kHz) and an isotropic FOV are needed (Supplementary Section 2.4). This allows for more efficient remapping onto a minimal number of rows, requiring less complexity in the fiber bundle. Although the bundle itself cannot be reconfigured after fabrication, sHAPR is highly versatile because it eliminates the need for mechanical components, pulsed lasers, and other hardware elements that could hinder compatibility with common imaging techniques, introduce distortions, or even limit the maximum imaging speed available. Furthermore, despite losses from the fiber bundle, sHAPR maximizes effective integration times by exposing all pixels simultaneously, enabling our system to maintain significantly higher SNR levels than similar high-speed imaging techniques (Supplementary Section 2.7).

Notably, our prototype setup can still be substantially improved, and future work may involve different aspects such as fiber production, sensor architecture, and image processing (Supplementary Section 2.5). In the current implementation, the $4 \times 4$-pixel area for each fiber projection results in a binning effect that reduces the effective number of pixels on the image compared to the camera sensor. Accordingly, this can be addressed by the implementation of smaller, tightly aligned fibers to yield considerable enhancements to the image resolution and field of view. The current two-fold loss from the fiber bundle can be greatly minimized by adding a fiber-coupled microlens array and an anti-reflection coating[101–103], further increasing the SNR. Meanwhile, the integration with different sensor architectures (e.g., linear arrays[104], intensified cameras[105,106]) or last-generation back-illuminated sCMOS sensors would improve the efficiency, available pixels, and imaging speed (>200 kHz, Supplementary Table 1)[26]. Finally, the combination of sHAPR with GPU-based computation would enable the real-time evaluation of more intricate tasks, which would be particularly interesting for ultra-fast image-activated operations like cell sorting[107] or event-driven imaging[108]. Consequently, we envision our system as a promising solution for a variety of high-speed imaging applications because of its versatility and user-friendly setup (Supplementary Section 2.3), paving the way for widespread use in biology, diagnostics, and pharmacology.

## Methods
### Imaging equipment
The system was constructed based on an epifluorescence microscope (Nikon Eclipse Ti2-U). Two different imaging objectives were used across the series of experiments: 100× 1.45NA objective lens (Nikon Plan Apochromat Lambda 100× Oil) and 40× 1.3NA objective lens (Nikon Plan Fluor 40× Oil). For fluorescence imaging, a 488 nm laser

was used for excitation (MPB Communications). A typical sCMOS camera operating in a reduced ROI mode was used as the ultra-high-speed imager (Hamamatsu ORCA-flash4.0 v2 C11440) and paired with a custom rectangle-to-line fiber bundle (Fiberoptics Technology Inc). The transmission of the fiber bundle is approximately 45%. The experimental implementation of the system is depicted in Supplementary Fig. S1.

### Alignment
The high-speed camera C1 was mounted on a three-axis stage, facing the fiber coupler output array. Because of the eight available rows for our camera, we chose to tune the magnification of CL to project each fiber onto 4-by-4-pixel regions. The position of the camera was adjusted by monitoring the real-time camera feed and adjusting the position of the camera until all fibers were within focus and centered as best as possible within 4-by-4-pixel regions. The pixel positions occupied by each fiber are then saved to form the initial location map. A MATLAB app was then created to overlay the initial location map image with the current camera image, thus providing a target alignment location (Supplementary Fig. S1e). For all future experiments, alignment was performed by manually tuning the position of the camera stage until the fibers coincided with their positions in the initial location map.

### Image reconstruction
To reconstruct the image, we employed a predefined mapping between the fiber's position on the FC-IA and its corresponding position in the FC-OA image, established through the calibration process detailed in Supplementary Section 2.2. For each individual fiber, we retained the FC-IA array index and the top-left pixel coordinates of its associated $4 \times 4$-pixel projection in the FC-OA image. The reconstruction method follows a straightforward rearrangement strategy: we sum the pixels within the $4 \times 4$ projection region for every fiber location in the FC-OA image. The result of this summation is then allocated to the respective index position of the fiber in the FC-IA, facilitating the reconstruction of the image plane. Optional settings control system normalization to correct for differences in transmission between fibers. First, an average of 100 frames at 10 ms exposure time with no illumination was taken and reconstructed, forming the background image. Then, the broadband white microscope lamp was turned on, and its intensity was adjusted until pixels were close to saturation; an average of 100 more frames were taken and reconstructed, forming the bright image. The background was subtracted from the bright image; on the resulting image, pixels were multiplied by gain factors to normalize the illumination across the entire image. These gain factors were saved as the normalization map. Reconstructions are then subject to background subtraction and multiplication by the normalization map when applicable to produce the final image (Supplementary Fig. S3). The reconstruction algorithm was implemented in MATLAB. Running on an Intel(R) Xeon(R) W-2145 CPU @ 3.70 GHz, each frame requires ~960 μs for processing, and eight frames are processed simultaneously via multi-core parallelization. 5,000,000 frames of experimental data were processed in 10 min. The open-source code used in this work is available as Supplementary Software. The most up-to-date version can be found at: https://github.com/ShuJiaLab/sHAPR.

### Microfluidic HeLa cell preparation
HeLa cells purchased from Sigma-Aldrich (ECACC, 93021013) were stained using Syto-16 (Thermo Fisher S7578). The cell culture medium was removed and replaced with the staining medium, which was made by dissolving Syto-16 into DMEM culture media to produce a final concentration of 4 μM. Cells were incubated in the staining medium for 45 min, then cells were washed twice using Hank's balanced salt solution (HBSS) without phenol red (Corning 21022CV). After HBSS was

removed, 1.5 mL of trypsin-EDTA (Thermo Fisher 25200056) was added to the dish for 1 min, gently swirled, and removed. The cell dish was placed inside the incubator at 37 °C for 3 min to detach the cells. Once the incubation was done, cells were resuspended into 3 mL of 4% PFA fixation buffer (16% PFA (Electron microscopy sciences 15710) with PFA:PBS:ultrapure-water in a 1:2:1 ratio) in a 5 mL vial at room temperature for 12 min. Cells were concentrated by centrifuging for 6 min at 800×$g$. Then the superficial solution was discarded, and cells were resuspended into 3 mL of clear PBS (Thermo Fisher 10010023). This washing step was repeated again, and cells were finally stored in 3 mL of PBS without phenol red for imaging.

### Microfluidic experimental procedure

Syto-16 HeLa cell solution prepared as previously described was flowed through a 30 μm wide channel on a purchased PMMA microfluidic chip (Fluidic 386; Chipshop). Flow-focusing channels on the chip were not used and were closed with plugs, restricting the flow to the central microfluidic channel of cross section 40 μm high × 300 μm wide. A syringe pump (Pump 11 Elite; Harvard Apparatus) was used to control the volumetric flow rate in order to achieve a desired estimated flow velocity, V, calculated using the equation:

$$V = \frac{Q}{A} \tag{1}$$

where Q is the volumetric flow rate, and A is the cross-sectional area of the channel. Combinations of different objectives (100×, 40×, 20×) and flow velocities (1, 0.8, 0.6, 0.4, 0.2, 0.1 m/s) were recorded. The 488 nm illumination laser power was set to 240 mW, and stroboscopic illumination was performed as described in Supplementary Section 3.1. For each test, the syringe pump was set to the desired volumetric flow rate and allowed to run for 5 s to reach a steady-state flow, after which 30,000 frames of high-speed video were captured at a framerate of 25,600 FPS. Subsequent cell tracking was performed using the ImageJ TrackMate plugin[109], as described in Supplementary Section 3.2 and Supplementary Fig. S4.

### Cardiomyocyte preparation

Cardiomyocytes were differentiated from IMR-90 IV hiPSCs (WiCell Research Institute) maintained in mTeSR1 media (Stem Cell Technologies) as described[110]. Briefly, cells were treated with 100 ng/mL recombinant human activin A (R&D Systems) in RPMI medium with 2% B27 insulin-free (RPMI/B27 insulin-free medium) on days 0. After 24 h, the medium was replaced with 10 ng/mL recombinant human bone morphogenic protein-4 (BMP4; R&D Systems) in RPMI/B27 insulin-free medium from day 1 to day 4. The medium was changed to RPMI medium with 2% B27 containing insulin (RPMI/B27 medium) on day 4. Cardiac spheres were generated on differentiation day 5[110]. Cells were dissociated with 0.25% trypsin/EDTA and seeded into AggreWell 400 plates (Stem Cell Technologies, #34415) at 1500 cells/microwell to allow cells to form cardiac spheres. Before cell seeding, plates with 1 mL/well of RPMI/B27 medium were centrifuged at 1000×$g$ to release trapped bubbles in microwells. To prevent cell death, the medium was supplemented with 10 μM of Rock inhibitor Y-27632 (Selleck Chemicals). Plates were centrifuged at 100×$g$ to compact single cells and then placed in an incubator. After 24 h, spheres were harvested to remove the Rock inhibitor and cultured with RPMI/B27 medium in the suspension dishes. Differentiated cells were maintained until the day of testing <30 days post differentiation. The medium was changed every 2 days. Immediately prior to imaging, cells were labeled with calcium indicators. In the whole-cell contraction experiment, 1 μM Fluo-4 (Thermo Fisher F14201) was used, and in the calcium spark experiment, 10 μM Oregon Green (Thermo Fisher O6807) was used. Cells were labeled with indicators for 45 minutes at 37 °C in a modified completed DMEM medium containing DMEM (Thermo Fisher

11965084) supplemented with 10% FBS (Corning 35011CV), 1% Gluta-MAX (Thermo Fisher 11140050), 1% MEM NEAA (Thermo Fisher 35050061), 1% Antibiotic Antimycotic Solution (Thermo Fisher 15240096), and 4 mg/mL AlbuMAX™ II Lipid-Rich BSA (Thermo Fisher 11021029). After labeling, cells were washed twice with HBSS (Corning 21021CV) and then incubated for 15 min in 1X normal Tyrode solution composed of 140 mM KCl (Sigma 793590), 5 mM NaCl (Sigma S9625), 1.0 mM MgCl$_2$ (Sigma 63069), 20 mM HEPES (Sigma H0887), 1.8 mM CaCl$_2$ (Sigma 21115), and 20 mM glucose (Sigma G7528-) adjusted to PH 7.4 using NaOH. Upon washing with HBSS three times, cells were further incubated in the completed DMEM medium for 15 min for recovery. Right before imaging, the completed DMEM was removed, and the cells were rinsed twice with HBSS without phenol red (21022CV, Corning), and 2 mL Fluobright DMEM (A1896701, Thermo Fisher) was added into the dish.

### Cardiomyocyte experimental procedure

Prepared culture dishes were placed inside a stage-top incubator (H301-K-Frame; OKOLAB) coupled to temperature (T-UNIT-BL-PLUS; OKOLAB), humidity (HM-ACTIVE; OKOLAB), and gas flow (CO2-Unit-BL; OKOLAB) controllers with a central interface (OKO-Touch; OKO-LAB). Before mounting the cells on the microscope, the incubator system was allowed to settle at 37 °C with a 5% CO$_2$ concentration and 95% relative humidity. Cells were illuminated using a 488 nm illumination laser set to 30–120 mW. Imaging was performed with a 100× objective. Spontaneous cardiomyocyte activity was captured at 1000 Hz, 9.35 kHz, or 25.6 kHz for up to 5 s per cell.

### Cardiomyocyte analysis

Videos were processed by performing background subtraction and filtering their pixel time series with a low pass filter with a cutoff frequency of 100 Hz. Then, baseline normalization was performed to generate F/F$_0$ data. To calculate key cardiomyocyte activity metrics, raw fluorescent pixel time series were exported to a standard electrophysiology analysis software (pCLAMP 10.6; Molecular Devices)[57]. Within the pCLAMP software, event detection was performed on each cardiomyocyte trace with a product of two exponentials as the fitting function (Supplementary Section 2.1, Fig. S2). Each fitting function was subsequently exported to MATLAB and solved for the relevant parameters such as t$_p$, F$_{max}$, and t$_r$.

### Neuron preparation

To prepare dissociated primary hippocampal neuron cultures, 18.5-day-old embryos were collected from timed-pregnant Sprague Dawley rats (Charles River Laboratories), and then hippocampi were dissected in ice-cold Hank's Balanced Salt Solution (HBSS). Hippocampi were pooled together, trypsinized for 15 min, briefly incubated in 20% fetal bovine serum, and then plated in glass-bottom 35-mm dishes (GWST-3522, WillCo Wells) coated with 100 μg/ml poly-L-lysine (P2636; Sigma-Aldrich) at a density of ~325,000 cells per dish[111]. Hippocampal neurons were cultured in Neurobasal medium (21103049; Gibco) supplemented with B27 (17504044; Gibco), penicillin/streptomycin (30002CI; Thermo Fisher), and GlutaMax (35050061; Gibco), and fed one-half volume of fresh growth medium every 7 days post-plating. Sex was not considered in this study. Animal care and use were conducted following National Institutes of Health guidelines, and procedures were approved by the Institutional Animal Care and Use Committee at Emory University.

Hippocampal neurons cultured for 21–28 days in vitro (DIV21-28) were used for Ca$^{2+}$ imaging experiments. Cells were labeled with 10 μM Fluo-4AM (F14201, Thermo Fisher) in Neurobasal medium for 45 min at 37 °C. After three washes with a recording solution (119 mM NaCl, 5 mM KCl, 20 mM HEPES, 2 mM CaCl$_2$, 2 mM MgCl$_2$, 6 g/L sucrose, pH 7.4), the cells were further incubated in the recording solution in a 37 °C CO$_2$ incubator for 30 min before imaging.

## Neuron imaging experimental setup

In order to depolarize neurons, a custom field excitation probe was constructed following the general design presented previously[76]. Details about the design and construction of this probe are further presented in Supplementary Section 5.1 and Fig. S6. A biological electronic stimulation unit (S88 Biological Stimulator; Grass Instruments) was coupled to the probe. In order to synchronize the electrical stimulation pulse, sample illumination, and video capture, the AOTF controller and high-speed camera were used as slave devices to the stimulator unit and connected via hardware trigger inputs.

## Neuron imaging experimental procedure

Neurons were placed in a stage-top incubator with environmental parameters: 37 °C, 5% $CO_2$, and 95% relative humidity. The excitation probe was placed into the culture dish, immersing the platinum electrode lines in recording solution 1 mm above the layer of neurons. Electronic excitation pulses were set to 30 V amplitude with a 300 ms duration. The 488 nm laser power was set to 30 mW. A 40× or 100× objective lens was used. Candidate healthy neurons were selected based on qualitative evaluation by eye using the microscope eyepiece. After identifying a candidate neuron, an excitation pulse was triggered using the biological stimulator unit, which also triggers sample illumination and high-speed video capture. 30,000 frames are acquired at a rate of 25,600 Hz. This process was repeated at least ten times per neuron to improve SNR via post-processing stacking. Afterward, we take a low-speed, high-resolution video of the same neuron as a reference. We switched to camera C2 and performed the same synchronized electrical stimulation sequence, but instead, we acquired batches of 100 images, each of 2048 × 2048 pixels at a rate of 100 Hz.

## Neuron analysis

For a given neuron, all captured high-speed excitations were then stacked to achieve a higher SNR, and each pixel was individually fit to a sigmoid function, which well-represents the rise of the fluorescent signal[112]. Finally, an isochronal map indicating the 50% rise time was generated. A detailed description of neuron image post-processing is presented in Supplementary Section 5.2 and Fig. S7.

## Statistics and reproducibility

Target results were replicated reliably in at least two experiments. Microfluidic results were replicated reliably in at least two independent imaging sessions for each experiment. Cardiomyocytes were recorded across at least three independent experiments. Several distinct excitation-contraction events (>4) were recorded for each cell ($n = 3$). Neurons were recorded across at least three independent experiments. For the neuron and cardiomyocyte experiments, cells could not be pre-screened for activity before recording due to the high-speed and remapping of sHAPR. Therefore, after reviewing images, neurons that presented no fluorescent activity, presented weakening activity during acquisition, or did not respond to stimulation were excluded. Cardiomyocytes that presented no fluorescence activity or presented weakening activity during acquisition were excluded.

## Reporting summary

Further information on research design is available in the Nature Portfolio Reporting Summary linked to this article.

## Data availability

Calibration target datasets shown in the current study are included in the Supplementary Software as example data. Additional imaging datasets from the current study are available under restricted access for the reason of large files sizes, access can be obtained from the corresponding author upon request. Requests will be fulfilled within 2 weeks. Source data are provided with this paper.

## Code availability

The core package for sHAPR is available as Supplementary Software. The code has been written in MATLAB (Mathworks) R2019a and has been tested with versions R2019a and R2021a. The software is available as a MATLAB app and source code. To use this code, install the app in MATLAB and follow the instructions included in the repository readme.txt. The alignment code is also provided. The latest version of the software can be found at: https://github.com/ShuJiaLab/sHAPR.

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

## Acknowledgements

We acknowledge the support of the National Institutes of Health grants R35GM124846 (to S.J.) and R01AA028527 (to C.X.), the National Science Foundation grants EFMA1830941 and DBI2145235 (to S.J.), the National Science Foundation Graduate Research Fellowship DGE-2039655 (to C.Z.), and the Spanish Ministry of Science "Ramón y Cajal" grant RYC2021-032084-I (to B.M.). We would like to thank Professor James Zheng and Dr. Kenneth Myers at Emory University for helping with the neuronal cultures and biological stimulation equipment.

## Author contributions

B.M. and S.J. conceived and designed the project. B.M. designed, constructed, and calibrated the imaging system with the help of S.R. B.M. and C.Z. developed the optical reassignment code. C.Z. designed and performed the stroboscopic illumination facilitated flow cytometry and neuron imaging experiments. W.L. and P.F. prepared and provided live cardiomyocyte samples. C.Z. performed the image analysis. C.X. inspected the research results. S.J. supervised the overall project. B.M., C.Z., and S.J. wrote the manuscript with input from all authors.

## Competing interests

The authors declare no competing interests.
