## [Peer Review File · Nature Communications]

High-Speed Optical Imaging with sCMOS Pixel ReassignmentREVIEWER COMMENTS

Reviewer #1 (Remarks to the Author):

The authors presented a high-speed acquisition technique to accelerate the framerate of the captured image using fiber optics. The authors show ample experimental results from different applications, such as flow cytometry, cardiomyocyte contraction, and neuronal calcium waves. The overall writing and description are lucid.

However, my main concern is that the scientific innovations in this paper are insufficient. Increasing the throughput by limiting the number of scanning rows in the CMOS image sensor is a rather standard protocol. The major contribution of this work is to use fiber optics to transform a regular FOV of the image to a few rows in the CMOS image sensor to better utilize the number of rows that need to be scanned. However, as far as I know, this is done by the commercial fiber optic bundle, and the CMOS image sensor the author employed is also a standard one. So I don't see a lot of scientific merit here.

For the experimental sections, from my understanding, most of them are already-known phenomena, which other equipment/methods can also obtain. If so, what are the specific reasons to adopt the proposed equipment/method from the author? Is there any critical information that can only be obtained through the proposed equipment?

Here are the other comments:

1. The authors state that their method "pushes the boundaries of current fluorescence imaging capabilities," but it would be more convincing if some quantitative comparisons to other existing methods were provided. For instance, how does sHAPR compare to other methods in terms of imaging speed, signal-to-noise ratio, resolution, and field of view?
2. The manuscript discusses the use of a custom fiber bundle for spatially rearranging the light from the imaging system but does not provide sufficient technical details. How exactly is the light rearranged? Is it a customized bundle?
3. And what is the attenuation of light intensity passing through the fiber bundle?
4. The authors mention that sHAPR methodology can be integrated into virtually any existing microscopy systems without requiring extensive platform modifications. However,

the manuscript lacks a detailed discussion on the potential limitations or challenges when integrating with different types of microscopy systems. This information would be useful for readers in assessing the versatility and practicality of the method.

5. In p. 4, line 95, why do the authors choose a pixel region of 4x4 for each output from the the FC-OA? Are there any special considerations? I would suspect that a smaller number would further increase the acquisition speed since fewer rows of the CMOS sensor would be occupied, right?

6. The authors should provide more details about the computationally efficient algorithm used to revert the acquired image back to its original undisturbed form. How much time does it take to process a single frame?

7. From my understanding, the FOV of the image of the proposed work is still limited because you cannot use the entire image sensor, similar to ref. 25 and 26, right? Please clarify this in the manuscript.

Reviewer #2 (Remarks to the Author):

Summary

This manuscript by Mandracchia and colleagues introduces the innovative 'sHAPR' technique, which utilizes a fiber optic bundle to transform a 2-dimensional image from its traditional image plane, where a camera would typically be situated, into a quasi-linear format. This transformation is specifically tailored to match the readout mechanism of contemporary sCMOS cameras. In CMOS camera chips, all pixels in a given column share the same analog-to-digital converter, leading to limitations in readout speeds primarily determined by the number of active rows rather than columns. Consequently, experimenters can achieve remarkably high frame rates, upwards of 25 kHz, but only within a highly constrained aspect ratio, such as an 8 x 2048 pixel region of interest. In this study, the authors employ a rectangular bundle of fiber optics (20 x 40) to relay the 2D image into a linearized (2 x 400) output. This output can be projected onto an 8-row region of a CMOS camera, enabling the highest attainable frame rates while retaining a more conventional field of view aspect ratio.

While fiber bundle relays have previously been used to map images from one spatial context to another, such as from a curved image surface to a flat image sensor, this study stands out by specifically mapping an extended two-dimensional plane to a 'quasi' one-dimensional line. I find the approach clever and innovative in its use of the anisotropic readout speeds of modern CMOS cameras.

Given my background in neurobiology and expertise in microscopy and camera technology, I can provide insights into the general strategy and cell-biology imaging aspects of this work. However, I may not be able to fully contextualize this paper specifically within the landscape of imaging flow cytometry and its current limitations and modern approaches.

Major Concerns

While this paper indeed showcases compelling applications of the presented approach, I believe there is room for a more extensive discussion regarding the design considerations, strengths, and notably, the limitations of this technique. The frame rates reported here (25kHz) are already achievable with an 8-pixel wide image. They've made those same rates possible now with a 20-pixel wide image. However, it's worth considering that some may approach this device with skepticism in its current form, perceiving it as a relatively complex and inflexible addition that compromises total field of view and light efficiency for a modest bump in aspect ratio uniformity; something whose use is perhaps limited to highly specialized cases. The inflexibility arises from the permanent reduction of what was once a 2048 x 2048 pixel device to a 20x40 pixel device.

To provide insight into my perspective as a reader of this paper and to open the door for the authors to offer further clarification or insights, I'd like to delineate my thought process – noting in advance that advances in camera choice and fiber bundle design could partially alleviate these concerns.

My primary focus is on the numbers presented and how they compare with what one could achieve without the added complexity of a fiber optic relay. Let's consider the initial case presented of a field of view of 20 x 40 pixels, with an effective pixel size of 1 μ m (100x

objective with 1.5x tube lens). This configuration corresponds to a 20 x 40 μm field of view, totaling 800 square microns.

To attain a similarly undersampled pixel size of 1 μm in a conventional setup, one could begin with a low magnification lens, such as a 40x/1.3 or 1.4 NA lens – resulting in an effective pixel size of 0.1625 μm (6.5 μm /40x). Subsequent demagnification of the image using a modified tube lens or the same coupling lens (CL) that that authors use (which, based on their reported methods, I estimate to be approximately 5.77x), would achieve an effective pixel size of 0.94 μm – very close to the effective pixel size in the first part of the manuscript.

To attain the full 25kHz frame rate of the camera, one could then restrict the observation to the same 8-row region on the camera. However, without the width limitations from the fiber bundle, we now have an effective field of view 8 x 2048 μm (16,834 square microns; contingent upon limitations of the microscope's field number). In essence, by using a lower mag lens without the fiber coupler, we encompass a 20-fold larger total sample area than the 800 square microns achieved with the fiber coupler – albeit at the expense of a significantly elongated aspect ratio. Importantly though, we've reduced the complexity, obviated the need for post processing, regained light efficiency lost to the fiber couple, and reestablished the flexibility of using the entire camera chip when maximal frame rates are not necessary.

Prospective users considering the adoption of this method may have a pivotal question in mind: what aspect ratio and how many total pixels are required for the application? If your objects are highly anisotropic, or if ~ 8 pixels along the short dimension suffice, then it would appear that you could attain similar performance without the added complexity. But as your sample approaches a mandatory aspect ratio of 1, where having 20 pixels instead of 8 along the shorter dimension becomes crucial, then this spatial mapping technique becomes more advantageous, and the trade-offs in complexity and efficiency become increasingly justifiable.

I am keen to hear the authors' insights on this line of reasoning, including any rebuttals

regarding aspects I might have inadvertently overlooked or misunderstood. For the benefit of other readers who might harbor similar inquiries, I believe it would be a valuable addition to the paper or its supplementary material to explicitly explore the scenarios in which the authors consider the fiber-coupled mapping as indispensable compared to what could have been accomplished solely with lenses. Specifically, it would be helpful to quantify the trade-offs between speed, aspect ratio, and total number of pixels in various designs.

Minor Concerns

A crucial factor in achieving high frame rates is the “achievable” signal-to-noise ratio at a given exposure time: the speed of your camera becomes irrelevant if you are constrained by the availability of photons. However, the paper doesn't touch upon the aspect of light collection efficiency. The input face of the fiber bundle presumably loses some amount of light (due to cladding and issues of fill-factor; in the supplement they state a silica core of 100 μm diameter, with a pitch of 150 μm). Therefore, some measurement of light loss resulting from the fiber relay would be a valuable addition, as it becomes an important part of the calculation for those considering adapting an existing conventional setup.

A brief statement on the cost and availability of these fiber bundles would be useful.

There is a push in the field of neurobiology towards imaging voltage transients instead of the secondary (and typically much slower) calcium transients – a task that has been much more difficult due to technological limitations in both sensors and detectors. While the authors did show the actual direction of propagation of calcium signals (an impressive feat), I believe many readers in the field would be most interested to use this technique to image voltage, rather than calcium.

It would be nice to see more discussion on the degrees of freedom for future advances. I believe it's important to focus on how the fiber and camera dimensions relate to each other with regards to constraining the achievable field of view. In the current implementation, the authors map each fiber onto a 4x4 pixel area. With a 2 x 400 fiber output, that means they would be using roughly 1600 of the 2048 available columns on the camera (80% of the

available width). So, even if someone purchased a fiber bundle with more than 20x40 fibers, they would be limited to a maximum number of fibers no greater than about half the width of the camera chip before having to compromise in speed again. Newer chips do have widths of 3200 pixels or more; but ultimately it would appear that there is a pretty hard cap here on how far this 2d-to-1d strategy can be taken in terms of field of view.

I believe the authors could make more concrete what could be possible in the very near future. For example, a recent camera (Teledyne Kinetix) can achieve 21 kHz with an ROI of 3200 x 64. Using the same 4x4 pixel-per-fiber as used in this paper, and an extended fiber bundle, that camera could attain a 128 x 100 pixel field of view at 21kHz. That is particularly impressive. I think a table exploring what could be achieved with various modern camera chips, along with the feasibility of extended fiber bundles, would be a nice addition.

Final Thoughts

I want to emphasize that I genuinely appreciate the strategy presented in this manuscript. The core concept is both sound and straightforward, and the authors have effectively demonstrated its potential to enhance numerous applications currently constrained by acquisition speed. It has potential for future development, and current limitations resulting from small fiber bundle size or outdated camera technology could be easily improved. I believe it represents an advance that should be shared with the scientific community, and I support the publication of this paper in some form. However, I would greatly value a more in-depth discussion of the trade-offs associated with employing a fixed fiber bundle and a direct comparison (or discussion) to an approach solely reliant on low magnification, relay lenses, and camera cropping. I suspect the authors have dedicated considerable thought to this comparison, and explicitly addressing it would further strengthen the paper.

Signed,

Talley Lambert

Reviewer #3 (Remarks to the Author):

Summary and general comments on the article:

The paper details a technique to optimize the speed of the data acquisition of 2D images for specific applications in microscopy where the dynamic of the biological processes is faster than the current sCMOS cameras.

The authors have used the technique to produce measurements in real biological systems using fast imaging. The technique has been experimentally tested on many systems and the data will be shared.

The paper is well organized and has a good division between the material used in the article and the supplemental material section.

The submission also has strong experimental evidence to support for the proposed technique.

I think there are aspects that need to be covered to support the novelty of the proposed technique. The general comments below try to give some framework for the discussion or details that should be added to the article for further consideration.

Specific comments to highlight the novelty of the technique:

In several fields of research, such as high energy physics, nuclear physics, etc, optical silicon sensors are used together with fiber optics to ease the construction of fast light-collecting systems. SiPM (for example) are single or multiple-channel devices that allow for single photon counting at a very high-speed rate with nano-second timing resolution or better. These devices, and other silicon devices, are used in different arrangements to improve the efficiency of light collection, ease the construction of the detector architecture, etc. At the same time, fibers are used to ease the optical coupling between different materials. Fiber focal planes are also used in astronomy, to collect the light at different points in the sky and project it in silicon imagers. A set of fibers is also used to image biological systems, like LOT.

The authors should provide more insight into the challenges of implementing other existing technologies to solve the problems of the particular field of the article. At least to mention how this system provides a different approach compared to similar strategies in other fields. Or the way existing strategies in other fields cannot be applied directly to this field.

The CMOS camera provides millions of channels on a single chip, so the readout speed is really limited to the total number of channels to attend. The rearrangement of the camera on a given portion of the CMOS sensor for faster readout using the fiber, is similar to changing the sensor design to read fast in a subarray of 20x40 matrix. What is the difference between doing ROIC compared to this approach? is it because of the extra time to access different rows during the sensor readout?

3) It seems that the technique is giving promising experimental results using high-speed rate. It would be good if the author could explain the limitations of increasing the number of pixels from 800 to 1 million pixels (for example) to differentiate it from the ROIC procedure that converges to the sensor readout time when all the pixels of the matrix are used.

4) In addition to the positional mapping for faster readout, does the fiber provide an extra advantage for collecting the light? compared to the sensor quantum efficiency? does it allow for optimized antireflecting coatings? curved surface? compared to having the system exposed to the surface of the sensor?

Response to Reviewers:

We thank the Reviewers for thoroughly examining the manuscript and providing useful comments. Here, we provide a point-by-point response letter in which we have provided detailed responses to each Reviewer's comments and made substantial revisions in our revised manuscript, as presented in the following.

Response to Reviewer #1

The authors presented a high-speed acquisition technique to accelerate the framerate of the captured image using fiber optics. The authors show ample experimental results from different applications, such as flow cytometry, cardiomyocyte contraction, and neuronal calcium waves. The overall writing and description are lucid. However, my main concern is that the scientific innovations in this paper are insufficient. Increasing the throughput by limiting the number of scanning rows in the CMOS image sensor is a rather standard protocol. The major contribution of this work is to use fiber optics to transform a regular FOV of the image to a few rows in the CMOS image sensor to better utilize the number of rows that need to be scanned. However, as far as I know, this is done by the commercial fiber optic bundle, and the CMOS image sensor the author employed is also a standard one. So I don't see a lot of scientific merit here.

For the experimental sections, from my understanding, most of them are already-known phenomena, which other equipment/methods can also obtain. If so, what are the specific reasons to adopt the proposed equipment/method from the author? Is there any critical information that can only be obtained through the proposed equipment?

Response: We thank the Reviewer for the constructive comments and would like to use this opportunity to elucidate the points of concern.

First, we would like to address the **scientific merit of sHAPR**. Indeed, the CMOS camera framerate can be increased by reducing the number of active scanning rows. However, this strategy is no longer practical for cell imaging when the reduced scanning rows lead to an over-narrowed field of view. In practice for cell biology, isotropic fields of view (FOVs) are generally preferred, and the active region of a CMOS sensor is typically reduced to a minimum of approximately 100 rows¹⁻⁴. This restricts acquisition speed up to ~ 2 kHz^{5,6} and corresponds to a single-cell FOV of ~ 10 - 20 μm or less when using a $60\times$ or higher-magnification objective lens.

Further reduction of the active rows will severely limit or prohibit the visibility of full cellular details. Although such narrow regions of interest (ROIs) can be leveraged in specific applications such as axon imaging⁷, scan-based imaging⁸, and industrial inspection⁹ (see new **Supplementary Table S1**, here reported as **Table R1**), in general, continued narrowing of the FOV in exchange for imaging speed is inadequate for cell research. As a result, despite the potential for higher spatial imaging speeds, there have not been commonly accessible wide-field solutions that can enable CMOS sensors to match their operational limits at narrower ROIs. In response to the demand, we present this novel sHAPR strategy utilizing 2D-to-1D transformation to efficiently reformat an image to exploit the anisotropic readout rates of common CMOS cameras. Thus, sHAPR represents a noteworthy improvement upon the state-of-the-art in that it is the first system in high-speed fluorescence microscopy that allows for wide-field distortion-free imaging of isotropic FOVs that can achieve frame rates well beyond 20 kHz.

In this context, we next would like to address the Reviewer's concerns regarding the **significance of the sHAPR instrumentation**, particularly the integration of standard CMOS sensors and fiber bundles. It is crucial to emphasize that incorporating the widely adopted components in sHAPR is a strategic decision aimed at overcoming the limitations imposed by more complex and costly high-speed imaging setups, which often involve cumbersome components (e.g., galvo mirrors, acousto-optic deflectors, high-speed signal generators) that can compromise versatility and induce unwanted image artifacts. Unlike these methods, sHAPR is distinguished by its

simplicity, modular design, and instrumentation, making it universally and advantageously compatible with a broad spectrum of microscopy platforms (see new **Supplementary Table S3**, here reported as **Table R2**).

Lastly, we would like to elaborate on the **usability and adoption of sHAPR**. sHAPR represents a paradigm shift by facilitating high-speed imaging via passive wide-field detection, distinct from the scanning methods prevalent in high-speed fluorescence imaging. This approach inherently maximizes effective integration times by exposing all pixels simultaneously—a fundamental advantage that cannot be matched by scanning-based methods. Our design, therefore, not only leverages but also amplifies the inherent capabilities of CMOS sensors, presenting a significant step forward in high-throughput imaging technology. The adaptability of sHAPR is particularly relevant in the context of user-customized microscopy applications, where the ability to integrate seamlessly into a diverse array of setups, such as *in vivo* platforms, animal rigs, and other auxiliary tools, is paramount. To this end, sHAPR does not require special illumination schemes and can be integrated into the detection path of any wide-field microscope. This quality affords compatibility with a variety of imaging techniques, ranging from light-sheet microscopy to super-resolution modalities like SIM and STORM. Future design could readily transform sHAPR into a snap-on accessory attached to the camera lens mount, reducing alignment needs for true drop-in functionality (see new **Supplementary Figure S8**, here reported as **Figure R1**).

Thus, we view sHAPR as a transformative advancement in high-speed microscopy, facilitating distortion-free CMOS-based imaging at frame rates exceeding 20 kHz. Its architectural simplicity and modular design underscore widespread compatibility across diverse microscopy platforms, establishing it as an accessible yet powerful solution for enabling high-speed imaging applications. We have provided additional context to highlight these key features of sHAPR in the Discussion section of the Main Text, along with additional supplementary content (see **Supplementary Figure S8**, **Supplementary Tables S1** and **S3**), and further differentiate our approach from others in the subsequent response.

Main text lines 250-263 | In this study, we employed image reformatting with flexible fiber optic pixel reassignment to leverage the fast line readout inherent to sCMOS sensors, overcoming the tradeoff of FOV anisotropy and speed. We note that sHAPR achieves maximal benefit over conventional camera cropping under certain conditions, namely, when the desired FOV is small and isotropic, which allows it to be efficiently reshaped onto the narrowest camera region (**Supplementary Section 2.4**). Within the ecosystem of high-speed imaging approaches, sHAPR distinguishes itself by facilitating continuous high-speed, distortion-free, and isotropic wide-field imaging at the microsecond scale using CMOS cameras through straightforward means (**Supplementary Table S2**). sHAPR eliminates the need for mechanical components, pulsed lasers, and other hardware elements that might impede compatibility with common imaging techniques and introduce distortions, whilst also maximizing effective integration times by exposing all pixels simultaneously as a widefield approach. Involving minimal alignment, we envision that a diverse range of researchers even without optical expertise can easily implement sHAPR to achieve high-speed imaging beyond 25kHz on various custom platforms (**Supplementary Section 2.3**), providing rich opportunities to study high-speed phenomena in IFC, cellular and molecular signaling, and other applications.

Table R1 (new Supplementary Table S1): Application requirements of isotropic and anisotropic FOVs

	Isotropic FOV	Anisotropic FOV
SPIM	Orthogonal Illumination (e.g., LSM)  The sample is expected to extend equally on both lateral dimensions. The optical resolution is the same in both directions and isotropic sampling is recommended.	Oblique Illumination (e.g., SCAPE)  One of the image dimensions corresponds approximately to the sample depth and is expected to have less extent than the other.
Scan Imaging	Flow Cytometry  Flow control with micrometric precision during the whole measurement is hard to obtain. Throughput is normally limited by both camera speed and FOV in the flow direction. A small number of pixels in the flow direction makes the system prone to motion artifacts and limits the throughput. Still, a small number of pixels in the direction orthogonal to the flow reduces the image resolution considerably.	Industrial Inspection  Scan speed can be controlled mechanically with high precision. Image resolution is limited by sampling. Reducing pixels in the scan direction allows the addition of more pixels in the other direction, thereby improving resolution in a cost-efficient way. Throughput is generally limited by factors other than camera speed.
Tracking	Unconstrained Diffusion (e.g., proteins on cell membrane)  Diffusion is expected to be isotropic unless some phenomenon occurs. Motion direction is unknown even in the case of non-Brownian diffusion.	Hindered Diffusion (e.g., proteins on filopodia)  Diffusion is constrained by the morphology of the imaged region. Motion is practically limited to one direction.
Neuron Imaging	Soma & Dendrites Imaging  Signal propagation is expected to be isotropic. All the region around the soma is part of the area of interest.	Imaging of Single Axon  Only a single axon is monitored. Signal propagation is essentially one-dimensional.

Figure R1 (new **Supplementary Figure S8**). Adaptability of sHAPR. **(a)** Detector-side sHAPR combined with arbitrary illumination schemes. **(b)** sHAPR integration with complex systems (diSPIM) as a bare CMOS replacement. **(c)** Fast switching between standard and sHAPR imaging modules. CL, coupling lens. **(d)** Possible drop-in sHAPR module with integrated optics.

Other Comments:

1. The authors state that their method "pushes the boundaries of current fluorescence imaging capabilities," but it would be more convincing if some quantitative comparisons to other existing methods were provided. For instance, how does sHAPR compare to other methods in terms of imaging speed, signal-to-noise ratio, resolution, and field of view?

Response: We thank the Reviewer for the opportunity to improve our manuscript. In light of this comment, we have added a new **Supplementary Table S3** in the revised supplementary text, here reported as **Table R2**, which offers a quantitative comparison with other major methodologies.

In recent years, numerous ultrafast imaging techniques have emerged, but many either do not support fluorescence imaging or offer only a limited number of frames. Despite this, a few methods have successfully achieved sub-millisecond fluorescence imaging without these drawbacks. For instance, the comb-scan method exemplified by FIRE (fluorescence imaging using radiofrequency-tagged emission) enables high-speed imaging by encoding fluorescence emission using different radio frequencies¹⁰. However, radiofrequency tagging increases the shot noise and is incompatible with light sources such as LEDs because it can be used only with coherent light. In addition, it requires a wavelength-dependent encoding of spatial information, which makes multicolor imaging very challenging. An alternative approach is point-scanning, exemplified by FACED (free-space angular-chirp-enhanced delay), This enables multi-color imaging by generating the time stretch of a pulsed laser without spectral encoding^{11,12}. Yet, both comb scanning and point scanning rely on pulsed lasers, increasing implementation costs and limiting compatibility with diverse imaging systems. Moreover, they are intrinsically mono-dimensional imaging techniques that necessitate scanning mirrors for 2D imaging. However, these mirrors reduce achievable frame rates for these techniques below 16 kHz (due to mechanical inertia), introduce an anisotropic resolution, and produce an image distortion that must be corrected in post-processing. A simpler solution could be the adoption of high-end, high-speed cameras. This reduces the complexity of the system but at the cost of having to work with lower framerates (below 13kHz) and highly asymmetric FOVs. As discussed in the previous answer, this last constraint poses severe limitations when observing biological phenomena and restricts actual applications to few use-case scenarios^{3,7} (see new **Supplementary Table S1**, here reported as **Table R1**).

As shown in the new **Supplementary Table S3**, here reported as **Table R2**, sHAPR considerably improves both acquisition speed and system simplicity. Significantly, the acquisition rate of sHAPR is not constrained by

mechanical limitations, allowing for potential advancements by integrating cutting-edge sensors, thereby suggesting a pathway for further performance enhancements in future iterations.

Finally, we have updated the **Supplementary Information** with a simplified SNR analysis. A full comparison of SNR across the different imaging approaches requires the consideration of multiple experimental and design factors, including optical losses, detector models, dark current, quantum efficiencies, and more parameters that are not readily provided and available. We offer a starting point for SNR considerations, noting that sHAPR possesses key advantages in long exposures and cross-talk-free noise.

Supplementary Text, Section 2.7 | We offer a simplified signal-to-noise analysis to compare sHAPR with other imaging approaches, acknowledging that a comprehensive comparative analysis will necessitate careful consideration of specific design factors, including system losses, detector type, quantum efficiencies, dark current, and noise factors, among others. As an initial assessment, sHAPR maximizes the signal-to-noise ratio (SNR) by employing high-quality sCMOS cameras. When operated with short exposure times or in a cooled low-noise regime, dark and spurious noise becomes negligible, making readout and shot noise the primary sources of noise. Consequently, the SNR expression can be simplified to:

$$SNR_{sCMOS}(p) = I_p / \sqrt{I_p + R^2},$$

where I_p is the number of photons collected at pixel p during the integration time, and R represents the readout noise. Nowadays, the readout component for sCMOS cameras is typically $R \sim 1e^-$ or less due to recent industry advancements¹³. Comparatively, EMCCD cameras exhibit negligible readout noise but are generally slower and introduce noise associated with the amplification process, represented by the noise factor F :

$$SNR_{EMCCD}(p) = I_p / \sqrt{F^2 I_p},$$

where $F^2 \approx 2$.

Both sCMOS and EMCCD cameras capture pixels in parallel, maximizing exposure time for all pixels increasing the overall SNR compared to serial acquisition methods like point-scanning confocal systems, which require dividing the total SNR by the number of pixels:

$$SNR_{point-scan}(p) = I_p / \sqrt{W \cdot H \cdot I_p},$$

where W and H are the numbers of pixel columns and rows, respectively.

Comb-scan methods partially address this limitation by recording pixel rows in parallel, increasing both imaging speed and integration time. However, frequency domain pixel mapping introduces shot noise cross-talk:

$$SNR_{comb-scan}(p) = I_p / \sqrt{2 \cdot H \cdot I_T},$$

where I_T is the mean of the total number of photons collected in scan line¹⁰. Therefore, the shot noise measured from a particular pixel depends on how many photons are measured at all the other pixels in the line, so a set of bright pixels in a line scan will lead to more uncertainty in the measurement of all the other pixels, no matter their brightness.

Thus, as a widefield approach, sHAPR gains a significant advantage over high-speed scanning approaches due to simultaneous exposure over all pixels, whilst avoiding channel cross-talk.

Table R2 (new Supplementary Table S3). Comparison of sHAPR with other high-speed fluorescence imaging methods.

Reference	Sampling (μm)	FOV (pixels)	Spatial framerate	Image distortion	Imaging Category	Detector	Complexity	Cost	Compatibility
sHAPR (this work)	1.0	20×40	25.6kHz	No	Widefield	sCMOS	Low	\$	High
Diebold et. al. - FIRE¹⁰	0.47 (x), >0.47 (y) †	256×256	4.4kHz	Yes	Comb-scan	PMT	Very High	\$\$\$	Low
Wu et. al. - FACED^{12,14,15}	0.82 (x), 0.35 (y)	80×1200	3kHz	Yes	Point scan	PMT	Moderate	\$\$	Low
Voleti et. al. - SCAPE 2.0³	1.41 (x) 1.07 (y) ‡ 0.86 (z)	640×148	12.7kHz	No	Lightsheet scan	High-speed Intensified sCMOS	High	\$\$	Moderate
Mikami et. al. - FDM confocal¹⁶	0.21 (x) 0.18 (y)	190×231	16kHz	Yes	Comb-scan	PMT	Very High	\$\$\$	Low
Popovic et. al.⁷	4	12×80	10kHz	No	Widefield	High-speed CCD	Low	\$	High

† In the scanning direction, sampling is not constant because of the sinusoidal deflection pattern of the resonant mirror. Notably, the sine correction algorithm used to remove such distortion significantly lowers image resolution.

‡ The authors perform oblique scanning, which requires de-skewing of the acquired volume to translate the image from the camera coordinate system to the global coordinate system.

2. The manuscript discusses the use of a custom fiber bundle for spatially rearranging the light from the imaging system but does not provide sufficient technical details. How exactly is the light rearranged? Is it a customized bundle?

Response: We apologize for the lack of clarity regarding the fiber bundle used in our study. This component is a custom design fabricated to our specifications by Fiberoptics Technology Inc. (Connecticut). It is an incoherent bundle manufactured without a fixed mapping of fibers between the two ends. As a result, we initially conducted mapping and calibration as depicted in the revised **Supplementary Figure S2**, here reported as **Figure R2**. This process ensures accurate identification of the corresponding ends for each fiber. This crucial mapping has been integrated into our reconstruction algorithm, which is pivotal for reconstructing the intermediate image from the fiber coupler output array (FC-OA). We have updated the **Supplementary Information (Section 2.1)** for better clarity:

Supplementary text, lines 114-124 | A custom-designed “Rectangle to Line” fiber bundle was purchased from Fiberoptics Technology Inc. The 0.5-m long bundle consists of 800 individual fibers arranged in a 20×40 rectangle on one side of the bundle and a 2×400 strip on the other end. Fibers are comprised of a pure fused silica core $100\pm 3\ \mu\text{m}$ in diameter, clad with fluorine-doped silica $110\pm 3\ \mu\text{m}$ in diameter, and finally coated with a polyimide coating, resulting in a total diameter of $125\pm 3\ \mu\text{m}$ per individual fiber. Fibers are composed of high OH silica and have a numerical aperture (NA) of 0.22 ± 0.02 . The pitch between fibers is $150\ \mu\text{m}$. Each end of the bundle was flat polished to a $0.3\text{-}\mu\text{m}$ tolerance. The fiber bundle and imaging system are shown in **Figure S1**. The bundle transmission was measured using a power sensor (Thorlabs S130C & PM100D) to measure incoming light in front of the FC-IA and output light after the FC-OA. The average transmission of the fiber bundle was 45% at 525nm. The fiber bundle was an incoherent bundle; thus, the exact correspondence of the fiber locations on each face was unknown and had to be measured as described in **Figure S2**.

Figure R2 (revised **Supplementary Figure S2**). Diagram of fiber mapping setup and resultant fiber mapping indices. **(a)** Fiber mapping measurement scheme. M, mirror; L, lens; BS, beam splitter; C, camera; FC-IA, fiber coupler input array; FC-OA, fiber coupler output array. Left callout, an image of fiber on the FC-IA illuminated by the laser, captured by C1. Scale bar: 0.3 mm. Right callout, an image of corresponding illuminated fiber ending

on FC-OA captured by C2. Scale bar: 0.2 mm. **(b)** Resultant positions of corresponding fibers on the FC-IA and FC-OA as measured by the measurement scheme. Mapping is saved and used in all reconstructions.

3. And what is the attenuation of light intensity passing through the fiber bundle?

Response: The light intensity attenuation is specific to each fiber within the bundle, yielding an overall bundle transmission rate of 45%. In light of this comment, we updated the **Supplementary Information (Section 2.1)**. Currently, our setup utilizes a standard, unmodified fiber bundle. However, we acknowledge that implementing anti-reflection coatings and integrating a microlens array to address fill-factor limitations could significantly enhance the overall coupling efficiency. These potential improvements are considerations for future optimizations and are further discussed in the new **Supplementary Information Section 2.5** for readers' better information.

Supplementary text, lines 121-122 | The bundle transmission was measured using a power sensor (Thorlabs S130C & PM100D) and determined to be 45% at 525 nm.

4. The authors mention that sHAPR methodology can be integrated into virtually any existing microscopy systems without requiring extensive platform modifications. However, the manuscript lacks a detailed discussion on the potential limitations or challenges when integrating with different types of microscopy systems. This information would be useful for readers in assessing the versatility and practicality of the method.

Response: We thank the Reviewer for this crucial observation. In response to this comment, our manuscript has been expanded to include a detailed discussion of best practices recommended for users integrating sHAPR with diverse microscopy systems. We expect the addition to guide users through the integration process and address potential challenges and how to navigate them effectively.

Supplementary Text, Section 2.3 | The integration of sHAPR in the detection path can drastically improve the imaging speed of virtually any standard sCMOS camera-based widefield system used for microscopy. Indeed, sHAPR is based on the rearrangement of two-dimensional images into monodimensional wavefronts. This is achieved entirely through passive optical waveguides without requiring any encoding of the incoming information. For this reason, the system can accept any bidimensional image independently from the sample illumination method employed. Thereby, sHAPR is compatible with both label-free and fluorescence imaging techniques such as bright-field imaging, digital holography, total internal reflection microscopy, spinning disk confocal, and selective plane illumination microscopy. Notably, sHAPR integration would be especially advantageous for those imaging techniques that require the acquisition of multiple frames to generate a single image. This includes label-free methods like tomographic phase microscopy and Fourier ptychography, as well as common super-resolution techniques such as single-molecule localization or structured illumination microscopy. However, successfully integrating sHAPR into a microscopy system demands attention to key parameters.

Calibration and alignment. Correct calibration of the fiber mapping and precise alignment of the linear output from the fiber bundle with the camera's pixel array are imperative. Failure to achieve proper alignment may result in signal loss in specific pixels or signal crosstalk in more severe cases. To circumvent these issues, it is vital for users to ensure that the surface of the fiber output remains parallel to the surface of the CMOS sensor and that all fiber images are precisely focused and aligned along the lines of pixels. On the bright side, all fibers are glued together, so it is not necessary to check the alignment of all fibers. Instead, alignment checks can be generally restricted to only a few reference fibers at the extremes of the array (Figure S1e).

Magnification. The pitch between fibers and fiber size introduces a binning effect in the projection of the image from the microscope onto the fiber bundle. Users may need to utilize a higher-magnification compared to a standard camera approach when using sHAPR to achieve the desired sampling, which can be achieved by

utilizing a higher magnification objective lens or introducing a magnifying relay between the fiber bundle and microscope.

Data acquisition. Optimizing data acquisition necessitates positioning the fiber output at the center of the imaging sensor. This not only minimizes photo response non-uniformity¹⁷ but, in many scientific cameras, also facilitates the utilization of twice the number of pixels as many of these cameras use two-line readout systems.

Light collection. Efficient light collection is paramount in high-speed imaging scenarios. Users are encouraged to fine-tune the coupling lenses to match the camera pixel size with the maximum value allowable for each experiment. Users may also consider introducing optics such as a microlens array at the fiber input to achieve better light collection and fill factor. Finally, adding image intensifiers would provide further improvements for light-starved applications.

Data storage. Another challenge associated with high-speed imaging is the substantial data generated, even when working with small ROIs. One potential approach to address this concern is to work with 8-bit images. While this lowers the camera's saturation threshold, this threshold is typically not encountered at high imaging speeds. Moreover, this adjustment can often increase the maximum achievable recording speed in many cases.

5. In p. 4, line 95, why do the authors choose a pixel region of 4x4 for each output from the the FC-OA? Are there any special considerations? I would suspect that a smaller number would further increase the acquisition speed since fewer rows of the CMOS sensor would be occupied, right?

Response: We thank the Reviewer for pointing out this critical experimental consideration.

The choice of a 4x4-pixel region in our research is tailored to our specific equipment, namely the sCMOS camera (Hamamatsu ORCA-flash4.0 v2 C11440) and the fiber bundle used. Our selected sCMOS camera imposes a minimum ROI of 8 rows through its software, while the fiber bundle outputs 2 rows of fibers. To optimize ease of alignment and maximize the sensor's dynamic range, our design allocates each fiber to cover a 4x4-pixel region, making full use of the available sensor rows. As pointed out by the Reviewer, a smaller projected region for each fiber could significantly enhance the acquisition speed or FOV of this method. Particularly with our camera model, reducing the projected area per fiber while utilizing a fiber bundle with a higher fiber count could substantially increase the pixel count. This adjustment has the potential to record a FOV of 128x128 pixels at a 25.6kHz acquisition rate, as outlined in **Table R3**. Alternatively, using a different camera sensor capable of smaller row counts could also expedite the acquisition process. Indeed, modern cameras offer the possibility of exceeding 200kHz and, in some cases, achieving rates as high as 1.6MHz (e.g., Teledyne Kinetix⁶, additional possible configurations described in **Supplementary Table S2**, here reported as **Table R4**), further enhancing the acquisition speed. We have updated the Main and Supplementary Texts to better reflect this.

Table R3. Examples of possible improved sHAPR configurations with higher fiber densities to fully utilize the active portion of the camera matrix, compared to the speed of standard region-of-interest selection of the same size FOV. Camera model: Hamamatsu ORCA-flash4.0 v2 C11440.

Camera pixels available	Fiber projected size	Total Fiber Count	Possible sHAPR FOV (px)	Framerate using equivalent cropped camera FOV (px)
8x2048	4x4	1024	32x32	6.4kHz
8x2048	2x2	4096	64x64	3.2kHz
8x2048	1x1	16384	128x128	1.6kHz

Main text lines 95-97 | To make full use of the available rows in the ROI, the magnification of the CL is finely tuned such that each fiber end at the FC-OA is projected to a 4x4 pixel on the camera sensor, easing alignment and increasing dynamic range (Fig1b).

Main text lines 267-270 | For instance, the implementation of smaller, tightly aligned fibers, each projected onto a smaller area of camera pixels, would yield considerable enhancements to the image resolution and field of view, whilst fiber bundle losses can be minimized by adding a fiber-coupled microlens array and an anti-reflection coating¹⁸⁻²⁰.

Supplementary text, lines 208-212 | Further reducing fiber-to-pixel projection by adjusting coupling lens magnification and increasing fiber count can enhance the field of view (FOV) or resolution for the same given sensor area (**Figure S10d**). The ultimate limit to this strategy would be a 1-to-1 fiber-to-pixel projection, enabling framerates on the order of 200kHz or higher with FOVs greater than 128×128 pixels depending on the camera model, explored in **Table S2**.

6. The authors should provide more details about the computationally efficient algorithm used to revert the acquired image back to its original undisturbed form. How much time does it take to process a single frame?

Response: In light of this comment, we have updated the **Methods** section with this information. In fact, during acquisition, the images coming from the microscope are reshaped, recorded by the camera, and streamed to the hard drive. Then, the original data shape is recovered in post-processing using a custom algorithm. This loads the raw image stacks and rearranges the positions of pixels according to the stored fiber mapping. Given that data are not compressed or altered in any way, this does not require computationally expensive operations such as domain transforms or spatial filters.

Main Text, lines 323-329 | To reconstruct the image, we employed a predefined mapping between the fiber's position on the FC-IA and its corresponding position in the FC-OA image, established through the calibration process detailed in **Supplementary Section 2.2**. For each individual fiber, we retained the FC-IA array index and the top-left pixel coordinates of its associated 4×4-pixel projection in the FC-OA image. The reconstruction method follows a straightforward rearrangement strategy: we sum the pixels within the 4×4 projection region for every fiber location in the FC-OA image. The result of this summation is then allocated to the respective index position of the fiber in the FC-IA, facilitating the reconstruction of the image plane.

Main Text, lines 338-342 | The reconstruction algorithm was implemented on MATLAB. Running on an Intel(R) Xeon(R) W-2145 CPU @ 3.70GHz, each frame requires approximately 960 μs for processing, and 8 frames are processed simultaneously via multi-core parallelization. 5,000,000 frames of experimental data were processed in 10 minutes. The open-source code used in this work is available as Supplementary Software. The most up-to-date version can be found at <https://github.com/ShujiaLab/sHAPR>.

7. From my understanding, the FOV of the image of the proposed work is still limited because you cannot use the entire image sensor, similar to ref. 25 and 26, right? Please clarify this in the manuscript.

Response: Indeed, the FOV of our work is ultimately limited by the number of pixels in the active rows, as we operate the camera in a cropped sensor mode. References 25 and 26 are examples of standard sCMOS cameras employed in fluorescence imaging, demonstrating that increased speeds of approximately 2kHz are achieved when reducing the imaging area to ~100 active rows. sHAPR not only leverages this underlying principle of CMOS sensors to increase imaging speed with cropped ROI but, more significantly, overcomes the previously intrinsic anisotropic FOV. For example, our current work demonstrates a FOV of 20×40 effective pixels projected onto an underlying cropped sensor region of 8×2048 at an acquisition speed of 25.6kHz. One could alternatively capture the same effective area as sHAPR by directly reducing the camera sensor to a 20×40-pixel region with a relay system to match the desired magnification. However, this would result in an acquisition speed of only up to 10.2kHz, a reduction of 60% as compared with sHAPR. Thus, both existing approaches do not utilize the full image sensor, and sHAPR reformats an image to occupy pixels more efficiently in column-parallel architecture CMOS cameras to maximize readout speed.

We have clarified that our approach advances the conventional strategy of using a reduced active area in the image sensor for speed improvement in the main manuscript. We have also added **Supplementary Figure S9** highlighting the trade-offs between standard sensor cropping (ROIC) and sHAPR to better illustrate the relationship of FOV, speed, and underlying pixel acquisition shape in the new **Supplementary Figure S9**, presented here as **Figure R4** (included in the response to **Reviewer 3**).

Main Text, lines 65-67 | Subsequently, this image is projected onto a linear region of the detector operating in a reduced readout mode, enabling high-speed recording at the microsecond scale (from ~25 to 250 kHz, depending on the sCMOS sensors).

Main Text, lines 93-96 | The dimensionally remapped image emerging from FC-OA is relayed through a coupling lens (CL) and captured by an sCMOS camera, which operates in a reduced readout mode at its narrowest allowable ROI of 8×2048 pixels, thus facilitating the acquisition of up to 25,600 frames per second (**Fig. 1a-iii**).

Response to Reviewer #2

This manuscript by Mandracchia and colleagues introduces the innovative 'sHAPR' technique, which utilizes a fiber optic bundle to transform a 2-dimensional image from its traditional image plane, where a camera would typically be situated, into a quasi-linear format. This transformation is specifically tailored to match the readout mechanism of contemporary sCMOS cameras. In CMOS camera chips, all pixels in a given column share the same analog-to-digital converter, leading to limitations in readout speeds primarily determined by the number of active rows rather than columns. Consequently, experimenters can achieve remarkably high frame rates, upwards of 25 kHz, but only within a highly constrained aspect ratio, such as an 8 x 2048 pixel region of interest. In this study, the authors employ a rectangular bundle of fiber optics (20 x 40) to relay the 2D image into a linearized (2 x 400) output. This output can be projected onto an 8-row region of a CMOS camera, enabling the highest attainable frame rates while retaining a more conventional field of view aspect ratio.

While fiber bundle relays have previously been used to map images from one spatial context to another, such as from a curved image surface to a flat image sensor, this study stands out by specifically mapping an extended two-dimensional plane to a 'quasi' one-dimensional line. I find the approach clever and innovative in its use of the anisotropic readout speeds of modern CMOS cameras.

Given my background in neurobiology and expertise in microscopy and camera technology, I can provide insights into the general strategy and cell-biology imaging aspects of this work. However, I may not be able to fully contextualize this paper specifically within the landscape of imaging flow cytometry and its current limitations and modern approaches.

Response: We thank the Reviewer for the positive comments on the innovation of sHAPR. We expect to clarify all the concerns raised by the Reviewer in our response below.

Major Concerns

While this paper indeed showcases compelling applications of the presented approach, I believe there is room for a more extensive discussion regarding the design considerations, strengths, and notably, the limitations of this technique. The frame rates reported here (25kHz) are already achievable with an 8-pixel wide image. They've made those same rates possible now with a 20-pixel wide image. However, it's worth considering that some may approach this device with skepticism in its current form, perceiving it as a relatively complex and inflexible addition that compromises total field of view and light efficiency for a modest bump in aspect ratio uniformity; something whose use is perhaps limited to highly specialized cases. The inflexibility arises from the permanent reduction of what was once a 2048 x 2048 pixel device to a 20x40 pixel device.

To provide insight into my perspective as a reader of this paper and to open the door for the authors to offer further clarification or insights, I'd like to delineate my thought process – noting in advance that advances in camera choice and fiber bundle design could partially alleviate these concerns.

My primary focus is on the numbers presented and how they compare with what one could achieve without the added complexity of a fiber optic relay. Let's consider the initial case presented of a field of view of 20 x 40 pixels, with an effective pixel size of 1 μ m (100x objective with 1.5x tube lens). This configuration corresponds to a 20 x 40 μ m field of view, totaling 800 square microns.

To attain a similarly undersampled pixel size of 1 μ m in a conventional setup, one could begin with a low magnification lens, such as a 40x/1.3 or 1.4 NA lens – resulting in an effective pixel size of 0.1625 μ m (6.5 μ m/40x). Subsequent demagnification of the image using a modified tube lens or the same coupling lens (CL) that the authors use (which, based on their reported methods, I estimate to be approximately 5.77x), would achieve an effective pixel size of 0.94 μ m – very close to the effective pixel size in the first part of the manuscript.

To attain the full 25kHz frame rate of the camera, one could then restrict the observation to the same 8-row region on the camera. However, without the width limitations from the fiber bundle, we now have an effective field of view $8 \times 2048 \mu\text{m}$ (16,834 square microns; contingent upon limitations of the microscope's field number). In essence, by using a lower mag lens without the fiber coupler, we encompass a 20-fold larger total sample area than the 800 square microns achieved with the fiber coupler – albeit at the expense of a significantly elongated aspect ratio. Importantly though, we've reduced the complexity, obviated the need for post processing, regained light efficiency lost to the fiber couple, and reestablished the flexibility of using the entire camera chip when maximal frame rates are not necessary.

Prospective users considering the adoption of this method may have a pivotal question in mind: what aspect ratio and how many total pixels are required for the application? If your objects are highly anisotropic, or if ~ 8 pixels along the short dimension suffice, then it would appear that you could attain similar performance without the added complexity. But as your sample approaches a mandatory aspect ratio of 1, where having 20 pixels instead of 8 along the shorter dimension becomes crucial, then this spatial mapping technique becomes more advantageous, and the trade-offs in complexity and efficiency become increasingly justifiable.

I am keen to hear the authors' insights on this line of reasoning, including any rebuttals regarding aspects I might have inadvertently overlooked or misunderstood. For the benefit of other readers who might harbor similar inquiries, I believe it would be a valuable addition to the paper or its supplementary material to explicitly explore the scenarios in which the authors consider the fiber-coupled mapping as indispensable compared to what could have been accomplished solely with lenses. Specifically, it would be helpful to quantify the trade-offs between speed, aspect ratio, and total number of pixels in various designs.

Response: We thank the Reviewer for providing insightful perspectives and pointing out the unique properties and usability of sHAPR.

First, we would like to elaborate briefly on the **sHAPR components** to address the Reviewer's concerns about readers' potential confusion. sHAPR presents a crucial development in high-speed imaging by enabling passive wide-field detection. Importantly, sHAPR mitigates the need for specialized illumination arrangements, facilitating its incorporation into the optical detection pathway of any standard wide-field microscope. This versatility enables compatibility across a spectrum of imaging modalities, extending from light-sheet microscopy to advanced super-resolution techniques such as SIM and STORM. Remarkably, the sHAPR conversion is neither permanent nor complex, and the standard imaging modality can be easily recovered with a minimum effort of adjustments. For example, a fast switch between standard and sHAPR modality can be achieved with a simple addition of relay lenses. Also, sHAPR can be feasibly implemented as an attachable module for the camera lens mount, minimizing alignment requisites and enabling straightforward 'add-on' utility. We illustrated these properties in the additional **Supplementary Figure S8**, presented here as **Figure R1** (see above in the response to Reviewer 1). In general, the pixel reassignment strategy relaxes the tradeoff between speed, usable FOV, and resolution compared to simple camera cropping. This comes at the expense of light collection due to the usage of fibers. Nonetheless, introducing a microlens array and antireflection coating at the fiber input can achieve better light collection and fill factor. Also, adding image intensifiers would provide further improvements for light-starved applications. We have added further discussion about these aspects and the best practices recommended for users integrating sHAPR in the new **Supplementary Text, Sections 2.3 and 2.5**. Moreover, it is worth noting that the quantum efficiency of our system, without any of the abovementioned improvements, is comparable to other high-speed imaging technologies, such as comb-scan or point-scan methods, with the benefit of not introducing image distortion and much higher SNR due to parallel acquisition (typically one order of magnitude, depending on the FOV). A detailed discussion about the SNR has been added in the new **Supplementary Text, Section 2.7**.

Second, we would like to highlight the importance of having an **isotropic FOV**. Generally, it is viable to downsize the camera sensor to a comparable 20×40-pixel area. However, this would result in a reduced acquisition speed of up to 10.2kHz, approximately 60% lower compared to sHAPR. As the Reviewer correctly states, by using a lower magnification lens without the fiber coupler, we could have a 20-fold larger FOV. However, an area with such a high aspect ratio turns out to be useful only in a very limited range of scenarios. In contrast, sHAPR maximizes the underlying architectural advantage of CMOS sensors to increase imaging speed via a cropped ROI. Significantly, sHAPR overcomes the limitations associated with anisotropic FOV that would constrain a majority of biological imaging needs. For readers' better clarity, we have revised the **Supplementary Information** with the new **Supplementary Table S1**, here reported as **Table R1**, emphasizing how anisotropic FOVs present limitations for most biological imaging applications. Indeed, isotropic FOVs are predominantly employed in biological imaging due to their uniform coverage.

In brief, one illustrative example where high aspect ratios are used can be found in specially configured selective plane illumination microscopy (SPIM). In such specific SPIM setups, constraints preclude positioning the illumination at a right angle to the detection optics, necessitating the use of an oblique light sheet³. Consequently, one dimension of the captured image will be essentially along the sample depth, which is typically less extensive than the lateral dimensions, leading to asymmetric FOVs. In contrast, standard implementations of SPIM generally utilize isotropic FOVs, maintaining uniformity across all dimensions. In addition, various cell biological imaging applications demand isotropic FOVs, for example, in the context of tracking dynamic biological processes where the direction of movement is not predetermined, such as protein diffusion across the cell membrane^{21,22} or within virological synapses²³, necessitating isotropic detection strategies. Of course, in specific scenarios like the diffusion in filopodia²⁴ or imaging discrete structures such as individual axons⁷, anisotropic FOVs, although less common, are still applicable.

Conversely, for applications where objects traverse a known trajectory, such as flow cytometry or industrial inspection, detectors with high aspect ratios like linear CMOS or time delay and integration (TDI) cameras are advantageous. They offer extended FOVs with reduced costs and exposure times, mitigating the risk of motion blur. Yet, these detectors demand precise synchronization and alignment, which, although typically manageable in industrial settings, can pose challenges in fluid dynamics-based systems like imaging flow cytometry due to potential instability in flow rates, leading to image distortion. Furthermore, within microfluidic platforms, the spatial constraints often result in a significant proportion of the imaging area, especially in the lateral direction, being underutilized while simultaneously reducing the system throughput due to the lack of pixels in the direction of flow. All these challenges may be substantially mitigated when using 2D cameras, albeit at the cost of longer exposure times and higher image blurring. As shown in the main manuscript, our system provides the best of both worlds, i.e., the stability of 2D imaging with the speed of linear detectors, which results in a throughput higher than in both cases.

Furthermore, we would also like to emphasize that the current configuration represents the prototype form. The present fiber bundle comprises 800 fibers, each encompassing a 4×4-pixel area upon the sensor array. Enhancing the fiber bundle with a more intricate and refined design would permit the harnessing of the full sensor pixel capacity. **Table R3** illustrates the performance of sHAPR as the density of the fiber coupler increases, with each fiber occupying fewer pixels while the entire available camera columns are used. The new **Supplementary Table S2**, presented as **Table R4**, illustrates potential enhancements achievable with state-of-the-art cameras and an improved sHAPR bundle.

Minor Concerns

1. A crucial factor in achieving high frame rates is the “achievable” signal-to-noise ratio at a given exposure time: the speed of your camera becomes irrelevant if you are constrained by the availability of photons. However, the

paper doesn't touch upon the aspect of light collection efficiency. The input face of the fiber bundle presumably loses some amount of light (due to cladding and issues of fill-factor; in the supplement they state a silica core of 100 μm diameter, with a pitch of 150 μm). Therefore, some measurement of light loss resulting from the fiber relay would be a valuable addition, as it becomes an important part of the calculation for those considering adapting an existing conventional setup.

Response: The attenuation varies individually for each fiber, with an overall bundle transmission of 45%. The fibers in this study are without specialized processing. Future improvements can greatly enhance coupling efficiency (for further discussion, please refer to our **response to Comment #5** below and **Supplementary Section 2.4**). Accordingly, we have updated the revised **Supplementary Information (Section 2.1)**.

Supplementary text, lines 121-122 | The bundle transmission was measured using a power sensor (Thorlabs S130C & PM100D) and determined to be 45% at 525 nm.

2. A brief statement on the cost and availability of these fiber bundles would be useful.

Response: The fiber bundle presented in this work was a custom design ordered from Fiberoptics Technology Inc. (Connecticut), costing \$4,275 and requiring an 8-week lead time. Whilst 2D-to-1D style fiber bundles have been previously presented in literature²⁵, they typically remain customized and are not readily available as off-the-shelf products. We have updated the **Supplementary Information (Section 2.1)** accordingly:

Supplementary text, lines 124-125 | In total, the fiber bundle cost \$4,275 and required an 8-week lead time; similar custom fiber arrays can be fulfilled by a variety of fiber optic manufacturers.

3. There is a push in the field of neurobiology towards imaging voltage transients instead of the secondary (and typically much slower) calcium transients – a task that has been much more difficult due to technological limitations in both sensors and detectors. While the authors did show the actual direction of propagation of calcium signals (an impressive feat), I believe many readers in the field would be most interested to use this technique to image voltage, rather than calcium.

Response: We thank the Reviewer for providing insightful guidance in the field. We acknowledge the significance of voltage imaging, especially as the higher temporal dynamics of voltage imaging enable the direct capture of high-speed electrical signals, including action potential and oscillations^{26,27}. Furthermore, non-depolarization-associated phenomena such as hyperpolarization or minor fluctuations in membrane potentials do not trigger calcium influx, and thus, voltage imaging is a natural choice for imaging such processes²⁰. Though we did not demonstrate voltage imaging in this work, we view it as a promising domain for future iterations of sHAPR. Here, we would also like to summarize the potential challenges for our consideration in this direction.

Considering the higher potential imaging speeds of sHAPR, the moderate fiber bundle loss, and relatively lower signal levels typical of voltage indicators²⁶⁻²⁸, this reduction of the photon budget requires refined instrumental choices, specialized voltage indicators, or both. To date, the fastest direct voltage imaging rate is 10kHz utilizing a high-speed CCD⁷, and other works have used sample averaging^{7,29} to overcome signal issues. However, as the signals of interest change, the latter may not be a viable strategy – for example, to image varying spike trains that cannot be consistently overlaid or in animals with evolving activity.

Thus, we anticipate extending the applicability of sHAPR for high-speed voltage imaging in low-light conditions in upcoming iterations of sHAPR. In practice, the ability of our system can be readily improved towards this task via hardware upgrades. Leveraging next-generation cameras with higher sensitivity and quantum efficiency, or even intensified and burst-mode cameras with hundred to thousand-fold amplification, presents a straightforward and compatible way to gain both low-light sensitivity and speed. The fiber bundle can also be improved in a variety of ways, including the addition of an anti-reflection coating or microlens arrays to enhance

the fill factor. Furthermore, the selection of specific voltage indicators (variants of dyes or genetically encoded) depending on the target application³⁰ can further help optimize the speed and signal responses. Voltage imaging is inherently demanding, requiring sampling rates surpassing 20kHz to fully capture high-speed phenomena^{7,31} paired with fast-responding voltage indicators possessing micro-second response times^{32,33}. To this end, our work already showcases some of the fastest direct wide-field fluorescence imaging of neurons, and we believe sHAPR is in a remarkable position to easily capitalize upon state-of-the-art hardware developments to elevate the approach to the next level and meet these demands. We further discuss advancements in the subsequent response (**Comment #5**). We are excited about the possibilities of this work and certainly plan to make voltage imaging a reality with future implementations of sHAPR.

Additional context towards voltage imaging has been included in the experimental results section:

Main text lines 228-243 | We expect the applications of our technology to extend beyond calcium imaging. High-speed voltage imaging has enabled direct capture of high-speed electrical signals using indicators with microsecond response times^{32,33}, opening the door to signaling phenomena that cannot be captured by calcium influx indicators^{26,27,34}. As such, it is increasingly applied to neural imaging. A clear challenge towards enabling this high-speed voltage imaging is the lower signal level of voltage indicators²⁶⁻²⁸ and low exposure times associated with rapid framerates, limiting current imaging studies to approximately 10kHz⁷ rates while also requiring trial averaging to increase SNR^{7,29}. As demands for capturing dynamic and evolving neuronal activity, including rapid oscillations and animal activity, increase, along with requirements for >20kHz acquisition speeds to fully sample the highest frequency component of APs³¹, sHAPR is well-positioned to quickly adapt to and capitalize on next-generation low-light imagers. New CMOS cameras with extraordinarily low readout noise, high QE, and even integrated image intensifiers have been developed and can be swapped into the sHAPR setup to boost the signal. Whilst current system losses in the fiber bundle and an outdated camera model limit the signal collection ability of our current implementation, future improvements are easily achievable (**Supplementary Section 2.5**), and sHAPR can pave a promising avenue toward further explorations of complex interactions, ranging from kinetics to biophysics and intracellular communications.

4. It would be nice to see more discussion on the degrees of freedom for future advances. I believe it's important to focus on how the fiber and camera dimensions relate to each other with regards to constraining the achievable field of view. In the current implementation, the authors map each fiber onto a 4x4 pixel area. With a 2 x 400 fiber output, that means they would be using roughly 1600 of the 2048 available columns on the camera (80% of the available width). So, even if someone purchased a fiber bundle with more than 20x40 fibers, they would be limited to a maximum number of fibers no greater than about half the width of the camera chip before having to compromise in speed again. Newer chips do have widths of 3200 pixels or more; but ultimately it would appear that there is a pretty hard cap here on how far this 2d-to-1d strategy can be taken in terms of field of view.

Response: In terms of the maximum FOV, indeed, our strategy is limited by the number of active pixels on the sensor at a given framerate. To clarify, the 4x4 fiber projection area was chosen due to our particular combination of camera and fiber bundle, but this is not a strict requirement, and smaller projections could be feasibly realized. Please refer to our relevant response to **Reviewer 1's Comment #5** for a more extensive discussion about this aspect.

We also agree that there is a maximum limit on what can be achieved with our strategy, along with certain conditions under which it performs best. Assuming a perfect fiber bundle, our approach would be limited by the number of active pixels, with the best performance gains over the standard ROI reduction achieved with isotropic FOVs, as observed by the Reviewer. **Supplementary Figure S9**, here presented as **Figure R4**, shows example scenarios for a variety of desired FOVs, assuming a fiber bundle is made to cover all active pixels. As

shown, when the target imaging region is a small, isotropic FOV that can be efficiently compressed into a few lines, sHAPR achieves both the maximum acceleration and the lowest complexity when compared to targeting more anisotropic or very large imaging FOVs. In light of these comments, we have updated the **Main Text** and also introduced an analysis of the scaling of sHAPR with respect to the aspect ratio in the **Supplementary Text (Section 2.4)**.

Supplementary text, Section 2.4 | sHAPR operates on CMOS sensors utilizing column-parallel architectures, in which the framerate is proportional to the number of active pixel rows H rather than total number of pixels:

$$\text{FPS} \propto \frac{1}{H}.$$

Rewritten, the imaging speed of a CMOS operating with subarray readout is dependent on the aspect ratio of the region $r = \frac{W}{H}$ and the total number of pixels $N = H \times W$, where W is the number of active pixel columns:

$$\text{FPS}(r, N) \propto \sqrt{\frac{r}{N}}.$$

The key advantage of sHAPR is that by reshaping the input image, it maximizes the effective aspect ratio at the camera plane (r') independently from the aspect ratio of actual FOV (r) so that $r' > r$ unless we use the entire CMOS sensor. This implies that the sHAPR method will always be faster than naïve subarray acquisition (ROIC), especially when imaging isotropic targets ($r = 1$):

$$\text{FPS}_{\text{sHAPR}}(r', N; r) = \sqrt{r'} \text{FPS}_{\text{ROIC}}(r, N).$$

Thus, the speed gain offered by sHAPR is maximized when the desired imaging FOV is highly isotropic, whilst the underlying camera acquisition region it is mapped to is highly anisotropic (**Figure S8**). In the non-ideal case where the fiber bundle fibers are mapped to a larger pixel region (like 4×4 in this work), the resultant expression is modified by the side length of the projection region p :

$$\text{FPS}_{\text{sHAPR}}(r', N; r) = \frac{\sqrt{r'}}{p} \text{FPS}_{\text{ROIC}}(r, N).$$

5. I believe the authors could make more concrete what could be possible in the very near future. For example, a recent camera (Teledyne Kinetix) can achieve 21 kHz with an ROI of 3200 x 64. Using the same 4x4 pixel-per-fiber as used in this paper, and an extended fiber bundle, that camera could attain a 128 x 100 pixel field of view at 21kHz. That is particularly impressive. I think a table exploring what could be achieved with various modern camera chips, along with the feasibility of extended fiber bundles, would be a nice addition.

Response: In light of this comment, we have included a new table in the **Supplementary Information** listing example configurations of fiber bundle designs combined with modern cameras (**Table S1**), along with **Supplementary Figure S10**, presented as **Figure R3**, which depicts various possible avenues for the improvement of speed, FOV, and light collection. For example, using the Teledyne Kinetix camera described, one could achieve up to 1.6MHz acquisition using a single row in the “speed” camera configuration or trade-off additional rows of the camera for an extended FOV, such as a 3200×64 sensor region proposed by the Reviewer. The influence of the extended fiber bundle is particularly noteworthy due to its potential to decrease the projected pixel area for each fiber when used in combination. In an optimal case where a fiber is matched to a single pixel, a 160×160-pixel FOV acquired at 200kHz is achievable with the Kinetix camera. A 160×160-pixel FOV at 100kHz is viable in an easier-to-achieve scenario of a 2×2 projection. In terms of the feasibility of these extended fiber bundles, we have reached out to various fiber optics vendors (MEISU, Fibertech Optica) to specifically inquire about large-format bundles (128×128 to 8×2048) and have been confirmed that such bundles are readily

manufacturable. Furthermore, large-format fiber reformatting bundles applied in spectroscopy have already been custom-built in the lab to reach similar array sizes (96×81^{35}), proving the feasibility of such fiber bundle upgrades even without the assistance of a specialized manufacturer. We have included a discussion about the potential hardware improvements in the new **Supplementary Text (Section 2.5)**.

Supplementary text, Section 2.5 | sHAPR can be readily upgraded with modern hardware. Generally, there are three main areas we foresee for future hardware advances, summarized in **Figure S10**: (1) Fiber bundle format, (2) Fiber bundle light collection optics, (3) Camera hardware.

Regarding the fiber bundle format, the current 800-fiber bundle, mapping to 4×4 pixel regions, occupies 80% of active pixels. Enhancements may first involve increasing fiber count for full sensor coverage. Further reducing fiber-to-pixel projection by adjusting coupling lens magnification and increasing fiber count can enhance the field of view (FOV) or resolution for the same given sensor area (**Figure S10d**). The ultimate limit to this strategy would be a 1-to-1 fiber-to-pixel projection, enabling framerates on the order of 200kHz or higher with FOVs greater than 128×128 pixels depending on the camera model, explored in **Table S2**. Fiber-optic manufacturing companies are capable of custom fiber bundles in such large formats (eg, 128×128 to 8×2048), and similar size arrays could also be hand-fabricated in lab³⁵. Additionally, the shape of the input array may be varied to achieve FOVs with varying anisotropy. Of course, fiber formats should also be selected with the camera hardware in mind, as other sensors may have different minimum row restrictions and pixel counts.

Fiber bundle light collection optics are crucial for increasing signal. Anti-reflection coatings can be applied onto fiber surfaces with deposition processes (**Figure S10b**) to decrease reflective loss by a few percent per surface ($\sim 2-4\%$). Furthermore, the fiber bundle fill factor is a major contributor to light loss, as only the core of the fiber bundle is receptive to light. This can be mitigated through the inclusion of a microlens array on the input side of the fiber bundle (**Figure S10c**), reflecting a similar approach taken in CMOS cameras to overcome pixel fill-factor issues³⁶.

Finally, we can leverage new cameras that offer combinations of extraordinary speed, high sensitivity, high quantum efficiency, and large imaging arrays (**Table S2**). Specialized options, such as new intensified sCMOS cameras with embedded image intensifiers, can enhance low-light performance (**Figure S10e**). sHAPR's compatibility extends even to EM-CCD cameras for burst imaging, utilizing their charge-shifting readout system in which rows are still exposed during the readout itself.

sHAPR provides a highly customizable approach, granting users a number of degrees of freedom in design. However, considerations for hardware compatibility are crucial. Matching fiber bundle output array shape to camera aspect ratios, accounting for different software limits, and adapting to varying pixel sizes with appropriate magnifying lenses are essential for optimal performance.

Table R4 (new Supplementary Table S2): Potential features of sHAPR when combined with state-of-the-art CMOS or CCD imagers.

Camera	Peak QE	Readout Noise	Active area	Fiber projection	FPS	Achievable sHAPR FOV	Required Fibers	Speed gain vs equal cropped FOV
Hamamatsu ORCA-flash4.0 v2 C11440 – This work	82%	1.6e ⁻	8×2048	4×4	25.6kHz**	20×40	800	2.5×
			8×2048	4×4	25.6kHz**	32×32	1024	4×
			8×2048	1×1*	25.6kHz**	128×128	16,384	16×
Teledyne Kinetix sCMOS (“Sensitivity” mode)	93%	1.2e ⁻	8×3200	4×4	35.4kHz	40×40	1600	5×
			1×3200	1×1	283.1kHz	50×64	3200	50×
			8×3200	1×1	35.4kHz	160×160	25,600	20×
Teledyne Kinetix sCMOS (“Speed” mode)	93%	2.0e ⁻	8×3200	4×4	200kHz	40×40	1600	5×
			1×3200	1×1	1.6MHz	50×64	3200	50×
			8×3200	1×1	200kHz	160×160	25,600	20×
HiCAM Fluor 2000	50%	24e ^{-***}	4×1920	4×4	243kHz	20×24	480	5×
			64×1920	4×4	33.5kHz	80×96	7680	1.25×
			4×1920	1×1	243kHz	80×96	7680	20×
Andor iXon 888 Life EMCCD	95%	<1.0e ^{-***}	1×1024	1×1	9.69kHz	32×32	1024	1×
			1024×1024†	1×1	1.6MHz†	32×32	1024	N/A

*Fibers are currently projected to a 4×4 pixel region, but future implementations could be used to project fibers onto a single pixel.

**ROI of camera is at the minimum allowed by camera software; the framerate is at a maximum for this particular camera model.

***Amplified cameras with adjustable gain levels

†This framerate is achievable based on the parallel-shifting or vertical clock functionality of the CCD camera, in which, during readout, the change on every pixel is shifted down by one row whilst still being exposed. This shift speed can be as low as 600ns, thus enabling a high-speed burst until all rows are read off the full sensor (1024 ‘frames’).

Figure R3 (new **Supplementary Figure S10**). Examples of possible sHAPR upgrades. **(a)** Current fiber bundle, exhibiting reflective loss and fill factor loss (yellow rays fall outside of core). **(b)** Anti-reflection coating can be applied through deposition processes (inset) and mitigates the reflective loss. **(c)** Micro-lens arrays can be fabricated through lithography (left inset) or purchased, anti-reflection coated (right inset), and bonded to fiber array to mitigate fill factor loss. **(d)** Smaller, densely packed fibers (right inset) with reduced coupling lens magnification (left inset) maximize pixel FOV. **(e)** Next-generation CMOS enables higher signal, imaging speed, and available pixels. Inset: intensified CMOS.

Final Thoughts

I want to emphasize that I genuinely appreciate the strategy presented in this manuscript. The core concept is both sound and straightforward, and the authors have effectively demonstrated its potential to enhance numerous applications currently constrained by acquisition speed. It has potential for future development, and current limitations resulting from small fiber bundle size or outdated camera technology could be easily improved. I believe it represents an advance that should be shared with the scientific community, and **I support the publication of this paper in some form.** However, I would greatly value a more in-depth discussion of the trade-offs associated with employing a fixed fiber bundle and a direct comparison (or discussion) to an approach solely reliant on low magnification, relay lenses, and camera cropping. I suspect the authors have dedicated considerable thought to this comparison, and explicitly addressing it would further strengthen the paper.

Response: We thank the Reviewer for the very positive recognition and support of this work. We also appreciate all the above insightful comments for helping strengthen the manuscript. We expect the above responses to address the Reviewer's remaining concerns about the work. Also, we have included more in-depth details of the advantages and trade-offs of sHAPR in the revised manuscript, which can be summarized as the following:

In general, sHAPR relaxes the trade-off between FOV, speed, and resolution typical of any imaging system. For instance, in scenarios with a fixed FOV and a desired imaging speed, sHAPR emerges as a superior choice, delivering enhanced resolution compared to classical methods utilizing relay lenses in conjunction with pure camera cropping. We demonstrate 25.6kHz imaging using a 20×40 array of fibers in our work. With a fiber pitch of 150 μm and magnification of the microscope (100× objective lens and extra 1.5× internal magnification), we achieve a 1- μm effective pixel size within a 20×40 μm FOV. In contrast, to maintain the same imaging speed and FOV with conventional cropping of a camera sensor, an ROI of just 8×16 pixels is required, resulting in a larger pixel size of 2.5 μm in comparison with sHAPR. Alternatively, if one were to maintain a sampling of 1 μm and utilize a cropped ROI of 20×40 pixels, the acquisition speed would be reduced by 60% to 10.2kHz in comparison to sHAPR. Thus, the sHAPR technology demonstrates enhanced performance relative to the conventional camera cropping method, offering improvements in speed, FOV, or resolution. However, the magnitude of these enhancements is contingent upon the selected aspect ratio, as previously delineated in **Comment #4** and elaborated upon in the **Supplementary Text, Section 2.4**. A direct comparison of camera cropping in various configurations has been added in **Supplementary Table S2**, presented here as **Table R3**.

Acknowledging the trade-offs integral to harnessing the performance of sHAPR, we highlight factors users may want to consider, along with strategies to mitigate potential downsides. While a fixed fiber bundle leads to reduced operational flexibility (but still adjustable by rearranging fiber patterns) in the aspect ratio of the FOV, users retain the ability to swap out the underlying camera sensor. Adjustments to the magnification of coupling lenses can compensate for varying pixel sizes or facilitate binning multiple fibers into a single pixel. Notably, sHAPR excels in isotropic imaging environments widespread in biology (discussed above in the main response), making it versatile even if the aspect ratio of the bundle itself is static. Losses attributed to the fiber bundle (approximately ~55%) can be mitigated through smart design choices, as discussed in **Comment #5**. Finally, we also note that there is an inherent binning effect in the projection of the image from the microscope onto the fiber bundle due to the pitch (~150 μm) between fibers, varying depending on the fiber size. This can also be readily adjusted to user-desired sampling by introducing a magnifying relay lens between the fiber bundle and microscope.

In light of this comment, we have expanded the **Main Text** and **Supplementary Information**:

Main text, lines 121-126 | This enhancement applies seamlessly to both bright-field and fluorescent targets, enabling sHAPR to surpass the limitations of traditional sensor cropping in standard sCMOS camera-based

systems. For instance, compared to selecting a sensor with a 20×40-pixel ROI, sHAPR showcases a framerate increase between 2.5 and 10 times, depending on the camera model. These performance gains become even more substantial with larger fiber bundles and more uniformly isotropic FOVs (**Supplementary Table S2, Supplementary Section 2.4-2.5**).

Main text, lines 252-255 | We note that sHAPR achieves maximal benefit over conventional camera cropping under certain conditions, namely, when the desired FOV is small and isotropic, which allows it to be efficiently reshaped onto the narrowest camera region (**Supplementary Section 2.4**).

Supplementary text, lines 164-168 | **Magnification.** The pitch between fibers introduces a binning effect in the projection of the image from the microscope onto the fiber bundle, varying depending on the fiber's size. Users may need to utilize a higher magnification compared to a standard camera approach when using sHAPR to achieve the desired sampling, which can be achieved by utilizing a higher magnification objective lens or introducing a magnifying relay between the fiber bundle and microscope.

Response to Reviewer #3

Summary and general comments on the article:

The paper details a technique to optimize the speed of the data acquisition of 2D images for specific applications in microscopy where the dynamic of the biological processes is faster than the current sCMOS cameras.

The authors have used the technique to produce measurements in real biological systems using fast imaging. The technique has been experimentally tested on many systems and the data will be shared.

The paper is well organized and has a good division between the material used in the article and the supplemental material section.

The submission also has strong experimental evidence to support for the proposed technique.

I think there are aspects that need to be covered to support the novelty of the proposed technique. The general comments below try to give some framework for the discussion or details that should be added to the article for further consideration.

Specific comments to highlight the novelty of the technique:

In several fields of research, such as high energy physics, nuclear physics, etc, optical silicon sensors are used together with fiber optics to ease the construction of fast light-collecting systems. SiPM (for example) are single or multiple-channel devices that allow for single photon counting at a very high-speed rate with nano-second timing resolution or better. These devices, and other silicon devices, are used in different arrangements to improve the efficiency of light collection, ease the construction of the detector architecture, etc. At the same time, fibers are used to ease the optical coupling between different materials. Fiber focal planes are also used in astronomy, to collect the light at different points in the sky and project it in silicon imagers. A set of fibers is also used to image biological systems, like LOT.

The authors should provide more insight into the challenges of implementing other existing technologies to solve the problems of the particular field of the article. At least to mention how this system provides a different approach compared to similar strategies in other fields. Or the way existing strategies in other fields cannot be applied directly to this field.

Response: We thank the Reviewer for the positive feedback and for providing us with constructive advice to improve the manuscript.

First, regarding the usage of fiber optics in other research areas, as the Reviewer has indicated, fibers have been commonly integrated with imaging components in various fields like nuclear physics, particle tracking, astronomy, medical imaging, and others. However, to our knowledge, many of these applications are aimed toward different goals, thus not fitting for use in optical microscopy. Silicon photomultipliers (SiPMs) have often been coupled to fibers, but these sensors are typically applied in different contexts. Radiation imaging and high-speed particle tracking often couple SiPM arrays with optical fibers and scintillating crystals³⁷⁻³⁹, but as these systems target radiation, the wavelengths and challenges associated with this domain make these systems inapplicable for fluorescence microscopy. Indeed, SiPMs can be utilized as fast detectors for biological microscopy⁴⁰. However, as single- or multi-channel devices, they require scanning to form a 2D image and thus are included in the discussion of scanning microscopy approaches presented previously.

Other common uses of fiber optics include images that do not provide any influence on imaging speed. For example, optic tapers are commonly used as magnifying image relays and are also often coupled to scintillators for X-ray or neutron imaging across fields including medicine and nuclear physics^{41,42}. In a similar manner, fiber optics bundles are widely utilized in biological imaging, including applications in LOT⁴³ and endoscopy⁴⁴, but

are similarly implemented as flexible image relays with no influence on imaging speed. Finally, some snapshot spectroscopy approaches in astronomy use fiber bundles in order to recover a spectrum for every pixel in the original image^{35,45-47}, rather than maximizing the CMOS throughput. In this regard, sHAPR represents a novel approach to the usage of fiber optics, enabling the image reformatting strategy in a new context toward optimizing the wide-field imaging speed of CMOS sensors.

In light of this comment, we have updated the main text to better reflect the difference between our approach and other high-speed fluorescence imaging strategies:

Main Text Lines 246-264 | In recent years, 2D scientific detectors have achieved single-photon sensitivity, but their recording speed remains constrained when compared to that of industry-grade cameras. However, all the attempts to bypass this shortcoming presented several limitations, such as the ability to provide only brief bursts of frames, short exposure dwell times, or the incompatibility with a wide range of imaging systems⁴⁸.

In this study, we employed image reformatting with flexible fiber optic pixel reassignment to leverage the fast line readout inherent to sCMOS sensors, overcoming the tradeoff of FOV anisotropy and speed. We note that sHAPR achieves maximal benefit over conventional camera cropping under certain conditions, namely, when the desired FOV is small and isotropic, which allows it to be efficiently reshaped onto the narrowest camera region (**Supplementary Section 2.4**). Within the ecosystem of high-speed imaging approaches, sHAPR distinguishes itself by facilitating continuous high-speed, distortion-free, and isotropic wide-field imaging at the microsecond scale using CMOS cameras through straightforward means (**Supplementary Table S2**). sHAPR eliminates the need for mechanical components, pulsed lasers, and other hardware elements that might impede compatibility with common imaging techniques and introduce distortions whilst also maximizing effective integration times by exposing all pixels simultaneously as a widefield approach. Involving minimal alignment, we envision that a diverse range of researchers, even without optical expertise, can easily implement sHAPR to achieve high-speed imaging beyond 25kHz on various custom platforms (**Supplementary Section 2.3**), providing rich opportunities to study high-speed phenomena in IFC, cellular, and molecular signaling, and other applications.

In regard to existing strategies within the field of high-speed biological imaging, please refer to our response to **Reviewer 1's Comment #1** and the new **Supplementary Table S3** (presented here as **Table R2**) for a detailed comparison with state-of-the-art strategies. In brief, sHAPR offers a highly compatible approach that can be adopted into a range of wide-field detection systems. It gains an inherent advantage in light efficiency over scanning methods, as it exposes all pixels simultaneously in contrast to exposing each pixel for only a fraction of the full image acquisition time. Finally, sHAPR merely requires the addition of a fiber bundle to a standard CMOS imaging setup. This accessibility makes our approach highly user-friendly and appealing to a broad user base, even those without specialized optical expertise. As a result, sHAPR represents a significant combination of high-speed performance and system versatility.

2) The CMOS camera provides millions of channels on a single chip, so the readout speed is really limited to the total number of channels to attend. The rearrangement of the camera on a given portion of the CMOS sensor for faster readout using the fiber, is similar to changing the sensor design to read fast in a subarray of 20x40 matrix. What is the difference between doing ROIC compared to this approach? is it because of the extra time to access different rows during the sensor readout?

Response: Indeed, the speed difference between classic ROIC and our approach is due to the additional readout time that extra rows incur. However, we want to clarify that for a typical CMOS camera, the readout speed is not limited by the total number of channels (pixels). The current architecture of CMOS cameras utilizes column-parallel architectures, which implement an analog-to-digital converter (ADC) per column of pixels. In comparison, a single ADC for the whole camera array runs into difficulties at high imaging speeds due to the extreme

clock speeds required for fast conversions, while one ADC-per-pixel reduces the fill-factor^{49,50}. As a result, modern CMOS cameras comprise the column-parallel architecture, and digitization, the bottleneck for the readout speed, is thus performed on all pixels in a row simultaneously. As a result, typical CMOS cameras that follow this architecture have a framerate dependent on the number of rows readout from the sensor (i.e., $\text{fps} \propto 1/H$) rather than the number of pixels (i.e., $\text{fps} \propto 1/(H \times W)$) like CCDs. This drives the motivation behind our work: the highest possible imaging speeds can be achieved when the number of active pixel rows is reduced to the allowable minimum, but doing so will reduce the FOV to a highly linear and anisotropic FOV. Though a CMOS camera may contain plentiful *pixels* in this regime (i.e., $8 \times 2048 = 16384$, $2 \times 3200 = 6400$), there is no way to utilize these pixels effectively for a majority of spatial imaging purposes of biological phenomena requiring isotropic FOVs, as the aspect ratio is fixed to a thin strip. sHAPR can utilize the available pixels more efficiently, allowing users to achieve both the speed and available pixels by salvaging this thin FOV to an isotropic one.

For example, to capture a 20×40-pixel region with sHAPR, 8 rows are enabled (the minimum rows allowed by our camera), resulting in a 25.6kHz framerate. In comparison, a direct ROIC to a 20×40-pixel subarray requires 2.5× more rows, which will result in a 60% lower framerate at ~10.2kHz. We would also like to emphasize that the gap between what can be achieved with ROIC compared to sHAPR will grow as the fiber bundle hardware develops and becomes more precise. At the moment, we project each fiber to a 4×4-pixel area and do not fully utilize the available area of the entire 8×2048 pixels. Consider a higher-density fiber array in which fibers could be projected to a 2×2 or even 1×1 pixel region on the sensor, as depicted in **Table R1**. In this scenario, with our current hardware, one could potentially achieve 25.6kHz in a 128×128-pixel FOV compared to ROIC, which would only achieve 1.6kHz for that same FOV.

We have added clarification of this column-parallel architecture to the main text:

Main text, lines 47-51 | sCMOS sensors permit enhancing the time resolution by selectively reducing the number of active pixel rows. Typically, with their high line readout rates surpassing 100 kHz, sCMOS cameras can attain around 1 kHz^{5,6} when tailored to a single-cell FOV (e.g., ~100×100 pixels, or tens of micrometers, **Fig. 1a**). However, owing to the column-parallel analog-digital converter architecture typical of CMOS sensors^{49,50}, framerates are dependent on the number of pixel rows rather than the total number of channels.

3) It seems that the technique is giving promising experimental results using high-speed rate. It would be good if the author could explain the limitations of increasing the number of pixels from 800 to 1 million pixels (for example) to differentiate it from the ROIC procedure that converges to the sensor readout time when all the pixels of the matrix are used.

Response: In light of this comment, we have added a more in-depth discussion on how the framerate between standard ROIC procedure and sHAPR relates in terms of desired FOV and active pixels in the **Supplementary Information**. In essence, sHAPR exploits the *irrelevant pixels* in a camera acquisition based on the desired image aspect ratio, with maximal accelerations and minimal bundle complexity achieved when imaging small and isotropic FOVs (for example, single-cell imaging of approximately up to ~100×100 pixels). A few different scenarios are shown in the new **Supplementary Figure S9**, presented here as **Figure R4**. sHAPR achieves maximal improvement when the desired FOV is isotropic and can be mapped onto a highly anisotropic region of the sensor. Eventually, as the entire camera sensor (e.g., over 1 million pixels as indicated by the Reviewer) is utilized, sHAPR will offer no improvement in terms of speed and, of course, has an added penalty in complexity due to the presence of the fiber bundle. Therefore, sHAPR is different from traditional ROIC because its image reshaping utilizes the available pixels in a more efficient fashion. On the other hand, many modern CMOS cameras can reach notably high speeds with active counts of rows < 8 (e.g., linear cameras), a regime in which ROIC can only form a high aspect ratio, anisotropic image, while sHAPR still allows for isotropic spatial imaging with substantially enhanced system accessibility.

Lastly, relevant to our response to **Reviewer 2's Comment #4**, we have updated the **Supplementary Text (Section 2.4)** detailing the scaling of sHAPR with different FOVs and aspect ratios.

Desired FOV	Traditional ROIC	sHAPR	Speedup Factor	Bundle Complexity
 Small, isotropic			High	Low
 Small, anisotropic			Medium	Medium
 Large			Low	High

Figure R4 (new **Supplementary Figure S9**). Examples of performance between traditional ROIC compared to sHAPR for different desired fields of view, assuming a fiber bundle fully utilizing the active pixel region. Solid blue: active pixels that are capturing the desired FOV. Dashed blue: pixels that can be read out with no speed penalty using traditional ROIC but are outside the desired FOV. Solid grey: inactive CMOS pixel rows.

4) In addition to the positional mapping for faster readout, does the fiber provide an extra advantage for collecting the light? compared to the sensor quantum efficiency? does it allow for optimized antireflecting coatings? curved surface? compared to having the system exposed to the surface of the sensor?

Response: We appreciate the Reviewer's attention to extended critical features of the utilization of a fiber bundle. Though our current prototype implementation does not provide all these benefits, we are excited about further possibilities and iterations of the sHAPR technology they can enable.

First, regarding light loss, current sCMOS cameras can achieve high quantum efficiencies (>80%), with state-of-the-art sensors exceeding 93%, built upon decades of design and manufacturing advancements. New sensors are made using backside illuminated (BSI) chip technology, in which the electronic gates and wiring elements of the chip are placed *underneath* the photoactive silicon pixel surface, eliminating the shadowing effects of the electronic wiring layer^{36,51}. With the silicon pixel surface directly exposed, the BSI technology has also enabled the application of anti-reflection coatings (ARCs) directly atop the silicon surface⁵¹, followed by the deposition of a microlens array onto this surface to mitigate the low fill factor of the pixels³⁶. In comparison, the additional component of the fiber bundle in sHAPR introduces extra loss avenues similar to those encountered during the development of CMOS sensors. Additional surfaces incur Fresnel reflection losses, and the fill factor of the fibers (the core area versus cladding and interstitial spacing) reduces the effective surface area for light collection.

Furthermore, though moderate, the fibers themselves possess losses associated with propagation distance and bending. In response, mirroring the development of the CMOS sensor, the addition of ARCs and microlens arrays to the fiber bundle can greatly reduce losses to perhaps a similar degree (~5%), which still remains an added loss to the overall system.

Nevertheless, we appreciate the Reviewer's insights on the potential advantages of fiber bundles and would like to elaborate on the possibilities to improve sHAPR, as depicted in **Figure R2**. With fiber bundles, though there will be loss induced by the fibers themselves, the usage of fiber bundles facilitates a higher efficiency for coupling to image intensifiers as opposed to relay lenses. Such intensified cameras have been effectively demonstrated toward enabling higher-throughput fluorescence microscopy³. Furthermore, the geometry of the input and output faces of the fiber bundle could be customized to achieve various curvatures. For example, curved polishing⁵² of the bundle may be utilized to reduce field curvature aberrations generated by the optics. The prototype system we present in this work does not yet contain any of these additional features, but we anticipate such additions to be powerful improvements for further implementations. In light of this comment, along with relevant **Reviewer 2's comment #5**, we have added new discussion content to the **Supplementary Text (Section 2.5)** and a new figure illustrating possible improvements (**Supplementary Figure S10**, presented here as **Figure R3**).

References

- 1 Otto, O., Gutsche, C., Kremer, F. & Keyser, U. F. Optical tweezers with 2.5 kHz bandwidth video detection for single-colloid electrophoresis. *Review of Scientific Instruments* **79** (2008).
- 2 Diekmann, R. *et al.* Characterization of an industry-grade CMOS camera well suited for single molecule localization microscopy—high performance super-resolution at low cost. *Scientific reports* **7**, 14425 (2017).
- 3 Voleti, V. *et al.* Real-time volumetric microscopy of in vivo dynamics and large-scale samples with SCAPE 2.0. *Nat Methods* **16**, 1054-1062 (2019). <https://doi.org:10.1038/s41592-019-0579-4>
- 4 Huang, F. *et al.* Video-rate nanoscopy using sCMOS camera--specific single-molecule localization algorithms. *Nature methods* **10**, 653-658 (2013).
- 5 Hamamatsu. <https://www.hamamatsu.com/eu/en/product/cameras/cmso-cameras/C13440-13420CU.html> (Hamamatsu Photonics K.K., Hamamatsu Photonics K.K., 2018).
- 6 Croucher, D. in *SPIE Exhibition Product Demonstrations*. 117162E.
- 7 Popovic, M. A., Foust, A. J., McCormick, D. A. & Zecevic, D. The spatio-temporal characteristics of action potential initiation in layer 5 pyramidal neurons: a voltage imaging study. *The Journal of physiology* **589**, 4167-4187 (2011).
- 8 Heng, X., Hsiung, F., Sadri, A. & Patt, P. Serial line scan encoding imaging cytometer for both adherent and suspended cells. *Anal Chem* **83**, 1587-1593 (2011). <https://doi.org:10.1021/ac102408g>
- 9 Line Scan Cameras, <<https://www.teledynedalsa.com/en/products/imaging/cameras/line-scan-cameras/>> (
- 10 Diebold, E. D., Buckley, B. W., Gossett, D. R. & Jalali, B. Digitally synthesized beat frequency multiplexing for sub-millisecond fluorescence microscopy. *Nature Photonics* **7**, 806-810 (2013). <https://doi.org:10.1038/nphoton.2013.245>
- 11 Lai, Q. T. K. *et al.* High-speed laser-scanning biological microscopy using FACED. *Nature Protocols* **16**, 4227-4264 (2021).
- 12 Wu, J.-L. *et al.* Ultrafast laser-scanning time-stretch imaging at visible wavelengths. *Light: Science & Applications* **6**, e16196-e16196 (2017).
- 13 Hamamatsu. (Hamamatsu Photonics K.K., Hamamatsu Photonics K.K., 2022).
- 14 Wu, J. *et al.* Kilohertz two-photon fluorescence microscopy imaging of neural activity in vivo. *Nature methods* **17**, 287-290 (2020).
- 15 Wu, J. *et al.* Multi-MHz laser-scanning single-cell fluorescence microscopy by spatiotemporally encoded virtual source array. *Biomedical Optics Express* **8**, 4160-4171 (2017).
- 16 Mikami, H. *et al.* Ultrafast confocal fluorescence microscopy beyond the fluorescence lifetime limit. *Optica* **5**, 117-126 (2018).
- 17 Mandracchia, B. *et al.* Fast and accurate sCMOS noise correction for fluorescence microscopy. *Nature communications* **11**, 94 (2020).
- 18 Warren-Smith, S. C., Dowler, A. & Ebdorff-Heidepriem, H. Soft-glass imaging microstructured optical fibers. *Opt Express* **26**, 33604-33612 (2018). <https://doi.org:10.1364/OE.26.033604>
- 19 Orth, A., Ploschner, M., Wilson, E. R., Maksymov, I. S. & Gibson, B. C. Optical fiber bundles: Ultra-slim light field imaging probes. *Sci Adv* **5**, eaav1555 (2019). <https://doi.org:10.1126/sciadv.aav1555>
- 20 Morova, B. *et al.* Fabrication and characterization of large numerical aperture, high-resolution optical fiber bundles based on high-contrast pairs of soft glasses for fluorescence imaging. *Opt Express* **27**, 9502-9515 (2019). <https://doi.org:10.1364/OE.27.009502>
- 21 Guo, M., Gelman, H. & Gruebele, M. Coupled protein diffusion and folding in the cell. *PLoS one* **9**, e113040 (2014).
- 22 Owen, D. M., Williamson, D., Rentero, C. & Gaus, K. Quantitative microscopy: protein dynamics and membrane organisation. *Traffic* **10**, 962-971 (2009).
- 23 Feldmann, J. & Schwartz, O. HIV-1 virological synapse: live imaging of transmission. *Viruses* **2**, 1666-1680 (2010).

- 24 Aratyn, Y. S., Schaus, T. E., Taylor, E. W. & Borisy, G. G. Intrinsic dynamic behavior of fascin in
filopodia. *Molecular biology of the cell* **18**, 3928-3940 (2007).
- 25 Schiza, M. V., Nelson, M. P., Myrick, M. & Angel, S. M. Use of a 2D to 1D dimension reduction fiber-
optic array for multiwavelength imaging sensors. *Applied Spectroscopy* **55**, 217-226 (2001).
- 26 Mollinedo-Gajate, I., Song, C. & Knöpfel, T. in *Concepts and Principles of Pharmacology: 100 Years of the
Handbook of Experimental Pharmacology* (eds James E. Barrett, Clive P. Page, & Martin C. Michel) 209-
229 (Springer International Publishing, 2019).
- 27 Antic, S. D., Empson, R. M. & Knöpfel, T. Voltage imaging to understand connections and functions
of neuronal circuits. *Journal of neurophysiology* **116**, 135-152 (2016).
- 28 Loew, L. M. in *Membrane Potential Imaging in the Nervous System and Heart* (eds Marco Canepari, Dejan
Zecevic, & Olivier Bernus) 27-53 (Springer International Publishing, 2015).
- 29 Hochbaum, D. R. *et al.* All-optical electrophysiology in mammalian neurons using engineered microbial
rhodopsins. *Nature methods* **11**, 825-833 (2014).
- 30 Aseyev, N., Ivanova, V., Balaban, P. & Nikitin, E. Current Practice in Using Voltage Imaging to Record
Fast Neuronal Activity: Successful Examples from Invertebrate to Mammalian Studies. *Biosensors* **13**,
648 (2023).
- 31 Yu, Y., Shu, Y. & McCormick, D. A. Cortical action potential backpropagation explains spike threshold
variability and rapid-onset kinetics. *Journal of Neuroscience* **28**, 7260-7272 (2008).
- 32 Baker, B. J. *et al.* Imaging brain activity with voltage-and calcium-sensitive dyes. *Cellular and molecular
neurobiology* **25**, 245-282 (2005).
- 33 Yu, Q., Wang, X. & Nie, L. Optical recording of brain functions based on voltage-sensitive dyes. *Chinese
Chemical Letters* **32**, 1879-1887 (2021).
- 34 Peterka, D. S., Takahashi, H. & Yuste, R. Imaging voltage in neurons. *Neuron* **69**, 9-21 (2011).
- 35 Wang, Y., Pawlowski, M. E. & Tkaczyk, T. S. in *Imaging, Manipulation, and Analysis of Biomolecules, Cells,
and Tissues IX*. 209-213 (SPIE).
- 36 Taylor, S. A. CCD and CMOS imaging array technologies: technology review. UK: *Xerox Research Centre
Europe*, 1-14 (1998).
- 37 García Díez, M. *et al.* Measurement of the bunch structure of a clinical proton beam using a SiPM
coupled to a plastic scintillator with an optical fiber. *Medical Physics* **50**, 3184-3190 (2023).
- 38 Beischer, B. *et al.* A high-resolution scintillating fiber tracker with silicon photomultiplier array readout.
*Nuclear Instruments and Methods in Physics Research Section A: Accelerators, Spectrometers, Detectors and Associated
Equipment* **622**, 542-554 (2010).
- 39 Risigo, F. *et al.* SiPM technology applied to radiation sensor development. *Nuclear Instruments and Methods
in Physics Research Section A: Accelerators, Spectrometers, Detectors and Associated Equipment* **607**, 75-77 (2009).
- 40 Modi, M. N., Daie, K., Turner, G. C. & Podgorski, K. Two-photon imaging with silicon
photomultipliers. *Optics express* **27**, 35830-35841 (2019).
- 41 Morgano, M. *et al.* Unlocking high spatial resolution in neutron imaging through an add-on fibre optics
taper. *Optics Express* **26**, 1809-1816 (2018).
- 42 Burger, R. J. & Greenberg, D. A. Fiber array optics for electronic imaging. *Electron Image Tubes and Image
Intensifiers II* **1449**, 174-185 (1991).
- 43 Burgess, S. A., Ratner, D., Chen, B. R. & Hillman, E. M. Fiber-optic and articulating arm
implementations of laminar optical tomography for clinical applications. *Biomedical optics express* **1**, 780-
790 (2010).
- 44 Elahi, S. F. & Wang, T. D. Future and advances in endoscopy. *Journal of biophotonics* **4**, 471-481 (2011).
- 45 Calcines, A., Harris, R. J., Haynes, R. & Haynes, D. in *Advances in Optical and Mechanical Technologies for
Telescopes and Instrumentation III*. 699-718 (SPIE).
- 46 Vanderriest, C. & Lemonnier, J.-P. in *Instrumentation for Ground-Based Optical Astronomy: Present and Future
The Ninth Santa Cruz Summer Workshop in Astronomy and Astrophysics, July 13–July 24, 1987, Lick
Observatory*. 304-310 (Springer).

- 47 Kriesel, J. *et al.* in *Algorithms and Technologies for Multispectral, Hyperspectral, and Ultraspectral Imagery XVIII*. 234-239 (SPIE).
- 48 Mikami, H., Gao, L. & Goda, K. Ultrafast optical imaging technology: principles and applications of emerging methods. *Nanophotonics* **5**, 497-509 (2016). <https://doi.org/10.1515/nanoph-2016-0026>
- 49 Lindgren, L. A new simultaneous multislope ADC architecture for array implementations. *IEEE Transactions on Circuits and Systems II: Express Briefs* **53**, 921-925 (2006).
- 50 Choubey, B., Mughal, W. & Gouveia, L. in *High Performance Silicon Imaging (Second Edition)* (ed Daniel Durini) 119-160 (Woodhead Publishing, 2020).
- 51 Lahav, A., Fenigstein, A., Strum, A. & Rizzolo, S. in *High Performance Silicon Imaging (Second Edition)* (ed Daniel Durini) 95-117 (Woodhead Publishing, 2020).
- 52 Ford, J. *et al.* in *Imaging Systems and Applications*. ITh1A. 4 (Optica Publishing Group).

REVIEWER COMMENTS

Reviewer #1 (Remarks to the Author):

The authors comprehensively addressed my concerns in the revision letter, which I deem the manuscript now is of high standard to be considered publication in the journal.

Reviewer #2 (Remarks to the Author):

I thank the authors for their extensive comments and revisions.

As mentioned in my first review, I do think the core concept presented here is sound, and the authors have proven it can be implemented. The additions to the supplementary data are welcome, and improve the paper.

My primary concerns and reservations are more editorial in nature. I feel that this manuscript, as written, and the author comments in the response to reviewers, tend a bit towards “salesy” language: attempting to persuade the reader of the merits of the approach rather than guiding the reader through its strengths and weaknesses of. A number of examples:

1. The authors write: “We introduce sHAPR, a high-speed acquisition technique that transcends the constraints of sCMOS cameras in capturing fast cellular and subcellular processes”. And later they call it “a paradigm that circumvents the conventional compromise for CMOS detectors”.

I would argue that it doesn’t “transcend” constraints or circumvent compromises: rather, it makes a fundamental compromise of total field of view for an isotropic, albeit highly restricted, aspect ratio. It’s a compromise that hasn’t previously been an option, which makes it helpful, but it’s not “better in most cases”, and we haven’t gotten anything “for free” here.

2. Signal collection and signal-to-noise ratio is a critical issue in high speed imaging –

perhaps the primary practical limitation. In the comments to reviewers, and in the revised supplement, the authors reveal that the bundle transmission is 45%. A 2-fold reduction in signal collection is a *major* compromise and should absolutely be mentioned in the main text and discussed. The authors, however, make it appear that sHAPR would be neutral to beneficial at best. In the section titled “Signal to noise analysis”, they completely omit this two-fold penalty in signal, but do state:

“sHAPR maximizes the signal-to-noise ratio (SNR) by employing high-quality sCMOS cameras”.

Here, I don't feel that the authors are making a good faith effort to discuss the merits of their approach with full disclosure of the compromises. I would prefer that they guide the reader to a more complete understanding, rather than highlight the good and brush over the bad.

3. The discussion of isotropic vs anisotropic FOV also feels like a slight indirection. They emphasize how “in cell biology, isotropic fields of view (FOVs) are generally preferred” (which is, of course, hard to argue). However, this isotropy is achieved at a *major* compromise of total number of pixels. For example: one could take a highly anisotropic FOV of 2000 x 100 pixels and turn it into an isotropic one by simply cutting off a 1900 x 100 ROI from one side, but that wouldn't feel like a “gain”.

4. In the discussion on system integration, they suggest that “sHAPR integration would be especially advantageous for ... common super-resolution techniques such as single-molecule localization or structured illumination microscopy”. I can tell you from experience that SIM is *incredibly* sensitive to anything that might modify the spatial arrangement of signal in the image (such as the fiber bundle). I would be extremely surprised if this device didn't make sim reconstruction nearly impossible. And localization microscopy is so sensitive to both issues of sampling and signal collection that the bump in maximum frame rate would likely be unusable due to more other more practical limitations.

In short, I feel like the statements around the broad applicability of this technique are highly speculative, and not grounded in any particular theoretical or practical demonstration.

None of this is not to imply that the core concept – the re-assignment of spatial information into a format more compatible with the row-wise readout of current sCMOS cameras – is not a clever and reasonable idea. The authors have demonstrated this proof of principle and that has value. But presenting it with a bit more explicit recognition of compromises would benefit the paper, rather than make it less exciting.

These may be editorial concerns, so should the paper be accepted for publication, perhaps the editorial team could take a closer look at the neutrality of the presentation.

On the authors' responses to reviewer #3 in the previous round of reviews:

Regarding point 1:

> The authors should provide more insight into the challenges of implementing other existing technologies to solve the problems of the particular field of the article.

First, a comment about my ability to evaluate this. I am not well versed in the other fields mentioned here. It would not surprise me if there are devices in other fields that bear similarities to this one, by using fiber optics to relay an image of sorts. I think the key question here is whether anyone has mapped a 2D array of fiber optics onto a fast 1D detector (of any sort), and that should be easy enough to determine (but I personally don't know of it being done in other fields). If not, I think it's novel enough, if so, that does indeed pose a problem.

> At least to mention how this system provides a different approach compared to similar strategies in other fields.

I agree in a that this would be nice, but it strikes me as bit more as commentary, perhaps even something in a review or a perspective piece.

In general, I take the authors at their word that usage of fibers in other fields, while perhaps common, hasn't quite touched on the application presented here. And if that's true, I think their response is sufficient. The only bit I would take a little issue with is their vision of ease of adoption:

"we envision that a diverse range of researchers, even without optical expertise, can easily implement sHAPR to achieve high-speed imaging beyond 25kHz on various custom platforms"

I'm not sure I share that vision. I think this is a clever, but relatively niche application that may find adoption among those at the bleeding edge of speed needs (those who specifically must have 20KHz+ speeds). The rest will likely wait for camera technology itself to reach higher speeds, before reaching for an intermediate image relay. But that is a digression.

In summary, I am satisfied with the authors response to this specific point – provided that it is indeed true that other fields don't commonly map a 2D fiber array onto a 1D line detector to take advantage of speed. And that is a provision I can't personally verify.

Regarding point 2:

The reviewer asked:

> The rearrangement of the camera on a given portion of the CMOS sensor for faster readout using the Fiber, is similar to changing the sensor design to read fast in a subarray of 20x40 matrix. What is the difference between doing ROIC compared to this approach? is it because of the extra time to access different rows during the sensor readout?

This question is a bit strangely worded, but it seems to be simply asking for an explanation of why readout of an 8 x 100 pixel ROI would be faster than that of a 20 x 40 pixel ROI on a

modern sCMOS. The answer is indeed as the authors say: on a CMOS camera, readout speed is limited only by the number of rows: the number of columns don't matter, and the total number of pixels doesn't matter (inasmuch as you don't change the number of rows involved in the readout). Their response to the reviewer is worded quite well I think. I also think that this point is fundamental (and commonly understood) enough that it doesn't require much additional detail in the text. The couple lines they add (47-51) are definitely sufficient.

I have one minor comment though: which is that I think the term "channels" (used by the reviewer) instead of "pixels" could be a bit confusing to many. I would use "pixels" and suggest slightly rephrase that sentence to remove the term "channels". Something like: However, owing to the column-parallel analog-digital converter architecture typical of CMOS sensors^{49,50}, framerates are dependent only on the number of rows – not the total number of pixels – in the active subarray.

Regarding point 3:

> It would be good if the author could explain the limitations of increasing the number of pixels from 800 to 1 million pixels (for example) to differentiate it from the ROIC procedure that converges to the sensor readout time when all the pixels of the matrix are used.

This question also makes me wonder whether the reviewer understood the basic mechanism of the speedup, and the limitations of CMOS cameras. I think the authors response is exactly right, and I commend them for pointing out that sHAPR actually gets less useful once the total number of pixels in the ROI can no longer be mapped onto a very thin (<8 row) strip of the camera. The new figure shows it nicely. This is a basic point and they address it well.

Regarding point 4:

> does the fiber provide an extra advantage for collecting the light? compared to the sensor quantum efficiency? does it allow for optimized antireflecting coatings? curved surface? compared to having the system exposed to the surface of the sensor?

I had a related question myself, but it was in the opposite direction. I assumed the addition of the fiber would hinder light collection. One still needs to project the image onto the camera, so we haven't circumvented the quantum efficiency or coatings of the sensor here. This can likely do nothing but decrease light throughput.

The ensuing discussion about back illuminated detectors, anti-reflection coatings and microlenses on the camera, intensified cameras, curved image planes and aberration reduction, all strike me as pretty much orthogonal to the important points at hand here.

The important point here (which the authors did address to some degree as well in response to one of my points) is that the fiber bundle reduces the light throughput roughly 55%. That's a big deal when trying to go fast (it's a 2-fold penalty in speed assuming signal-to-noise ratio is held constant). If I recall correctly, the main text and/or supplement do now mention this, but I still think the authors downplay the detrimental impact of the addition of a lossy optical element into a system where speed is critical.

In any case, with respect to the reviewer here. They've answered it just fine. The answer is no, not at all :)

Response to Reviewers:

We thank the Reviewers for thoroughly examining the revised manuscript and providing positive feedback. Here, we provide a point-by-point response letter in which we have provided detailed responses to each Reviewer's comments and made corresponding revisions in our revised manuscript, as presented in the following.

Response to Reviewer #1

The authors comprehensively addressed my concerns in the revision letter, which I deem the manuscript now is of high standard to be considered publication in the journal.

Response: We thank the Reviewer for considering the manuscript for publication in the journal.

Response to Reviewer #2

I thank the authors for their extensive comments and revisions.

As mentioned in my first review, I do think the core concept presented here is sound, and the authors have proven it can be implemented. The additions to the supplementary data are welcome, and improve the paper.

My primary concerns and reservations are more editorial in nature. I feel that this manuscript, as written, and the author comments in the response to reviewers, tend a bit towards "salesy" language: attempting to persuade the reader of the merits of the approach rather than guiding the reader through its strengths and weaknesses of. A number of examples:

Response: We understand the Reviewer's concern about the neutrality of the presentation and expect our further amendments to the manuscript to fully clarify these concerns.

1. The authors write: "We introduce sHAPR, a high-speed acquisition technique that transcends the constraints of sCMOS cameras in capturing fast cellular and subcellular processes". And later they call it "a paradigm that circumvents the conventional compromise for CMOS detectors".

I would argue that it doesn't "transcend" constraints or circumvent compromises: rather, it makes a fundamental compromise of total field of view for an isotropic, albeit highly restricted, aspect ratio. It's a compromise that hasn't previously been an option, which makes it helpful, but it's not "better in most cases", and we haven't gotten anything "for free" here.

Response: We agree with the Reviewer's perspective and apologize for any implications of breaking the boundaries of these fundamental tradeoffs. As the reviewer aptly noted, sHAPR opens a new choice to tradeoffs that were not available before by considering the CMOS architecture.

Specifically, we intended to reference the "conventional compromise" between FOV isotropy and speed, augmented by CMOS-imposed limitations. We also recognize that our technique is ultimately constrained by a fundamental trade-off encompassing total FOV, speed, and aspect ratio. Given these considerations, we have revised the manuscript language to more accurately reflect the capabilities of sHAPR.

Main Text, lines 15-16 | Here, we introduce sHAPR, a high-speed acquisition technique that leverages the operating principles of sCMOS cameras to capture fast cellular and subcellular processes.

Main Text, lines 61-63 | To overcome these limitations, we present sCMOS high-speed acquisition with pixel reassignment (sHAPR). We introduce an integrated optical and computation approach (**Fig. 1a**) that maximizes the sensor framerate for small and isotropic FOVs through careful consideration of CMOS detector architectures.

2. Signal collection and signal-to-noise ratio is a critical issue in high speed imaging – perhaps the primary practical limitation. In the comments to reviewers, and in the revised supplement, the authors reveal that the bundle transmission is 45%. A 2-fold reduction in signal collection is a *major* compromise and should

absolutely be mentioned in the main text and discussed. The authors, however, make it appear that sHAPR would be neutral to beneficial at best. In the section titled “Signal to noise analysis”, they completely omit this two-fold penalty in signal, but do state:

“sHAPR maximizes the signal-to-noise ratio (SNR) by employing high-quality sCMOS cameras”.

Here, I don't feel that the authors are making a good faith effort to discuss the merits of their approach with full disclosure of the compromises. I would prefer that they guide the reader to a more complete understanding, rather than highlight the good and brush over the bad.

Response: We agree with and appreciate the Reviewer's feedback. We have added information and discussion about the bundle transmission to multiple areas in the main text to offer better information to the readers, and the SNR analysis in the **Supplementary Text** now considers quantum efficiency. It is still evident that sHAPR provides benefits in SNR compared to high-speed scanning approaches, but these changes should make the loss of the bundle a clear factor.

Main text, lines 262-264 | Furthermore, despite losses from the fiber bundle, sHAPR maximizes effective integration times by exposing all pixels simultaneously, which enables our system to maintain a significantly higher SNR level compared to similar high-speed imaging techniques (**Supplementary Section 2.7**).

Main text, lines 271-272 | The current two-fold loss from the fiber bundle can be greatly minimized by adding a fiber-coupled micro-lens array and an anti-reflection coating, further increasing the SNR

Main text, lines 311-312 | The transmission of the fiber bundle is approximately 45%.

In **Supplementary Section 2.7**, we have updated the equations to include sensor quantum efficiency, Q_E . Also, we updated the last paragraph to include a better comparison between the SNR of different imaging modalities, taking the quantum efficiency into account:

Supplementary text, section 2.7, lines 302-313 | As shown by the equations above, high-speed imaging techniques must fundamentally trade off SNR for speed compared to conventional sCMOS or EMCCD cameras. Thus, while high-speed scanning methods can achieve significantly higher frame rates than EMCCD cameras, they have to sacrifice photon efficiency, resulting in significantly reduced SNR. Interestingly, the sHAPR technique maintains superior effective quantum efficiency (photon collection) than both scanning approaches despite the light transmission loss inherent in the current implementation. Furthermore, being a widefield approach, sHAPR has a notable advantage over scanning methods by allowing simultaneous exposure across all pixels and avoiding channel cross-talk. Consequently, sHAPR can offer even faster frame rates than both point- and comb-scan techniques and attain higher SNR levels closer to those of EMCCDs (**Figure S11**, see below). Additionally, as discussed in **Supplementary Section 2.5**, optimizing the fiber bundle design holds promise for mitigating quantum efficiency losses, potentially reaching or even surpassing the SNR values of regular EMCCDs.

Supplementary Figure S11. SNR yielded by different high-speed imaging techniques. Here, we plotted the SNR response of an EMCCD (gray), sHAPR (red), point-scan (blue), and comb-scan (yellow) according to the equations reported in **Supplementary Section 2.7**. For quantum efficiency, we considered 93% for the EMCCD, 37% for sHAPR, and 30% for both point- and comb-scan methods (as previously reported¹⁻³). Moreover, for scanning techniques, we considered a scanning area of 20×20 pixels (bigger areas yield lower SNR). Finally, for the comb scan, we considered a uniform intensity distribution.

Main text:

3. The discussion of isotropic vs anisotropic FOV also feels like a slight indirection. They emphasize how “in cell biology, isotropic fields of view (FOVs) are generally preferred” (which is, of course, hard to argue). However, this isotropy is achieved at a *major* compromise of total number of pixels. For example: one could take a highly anisotropic FOV of 2000 x 100 pixels and turn it into an isotropic one by simply cutting off a 1900 x 100 ROI from one side, but that wouldn't feel like a “gain”.

Response: While the Reviewer's observations hold true, we underscore the importance of isotropy in the context that sHAPR aims to benefit from. Typically, users seeking higher imaging speeds with a bare CMOS sensor can crop the active area to achieve a smaller FOV. As the Reviewer rightly points out, in many cases, the number of pixels remains high enough that the anisotropy of the FOV poses minimal impact on imaging tasks. However, to attain speeds comparable to that of sHAPR, users must significantly reduce the FOV in one direction to a few pixels (typically, ≤ 8). In such instances, while there may theoretically be a sufficient total number of pixels, the anisotropy of the FOV greatly affects the imaging capability, rendering it impractical in many scenarios.

In this context, sHAPR presents the option to maintain the same imaging speed while reorganizing the pixels in a more isotropic and practical manner. This is demonstrated by achieving a framerate of 25.6kHz with a 20×40 pixel FOV, a specification otherwise unattainable for the sensor. Hence, our statement focused on introducing an option for end users requiring high speeds, allowing them to customize the imaging FOV to a beneficial arrangement.

However, we acknowledge that the total FOV is compromised with the current approach due to the binning effect of the fibers, in which each fiber is captured by a larger 4×4 pixel region on the sensor. In this sense, we agree with the Reviewer that we should better present the incurred compromises. As such, we have revised the **Discussion** section and **Supplementary Material** to redirect the focus from isotropy to pixel reduction.

Main text, lines 253 | Deletion of “overcoming the tradeoff of FOV anisotropy and speed.”

Main text, lines 256-259 | Compared to conventional camera cropping, sHAPR offers a significant improvement when very high speeds (> 20 kHz) and an isotropic FOV are needed (**Supplementary Section 2.4**). This allows for more efficient remapping onto a minimal number of rows, requiring less complexity in the fiber bundle.

Main text, lines 268-269 | In the current implementation, the 4×4 pixel area for each fiber projection results in a binning effect that reduces the effective number of pixels on the image compared to the camera sensor.

Supplementary text, section 2.4 (This supplementary section has been adjusted to make the fiber binning effect clear):

However, the current implementation under-samples the imaging plane due to the binning effect of the fibers in the bundle, in which a single fiber is projected to a 4×4 pixel region. Thus, the number of pixels used in the camera sensor N is related to the number of fibers N_f by:

$$N = N_f \cdot p^2$$

Where p is the side length of the isotropic projection region in pixels. The acceleration factor is now:

$$f = \frac{FPS_{sHAPR}(r, N)}{FPS_{ROIC}(r', N_f)} = \frac{1}{p} \sqrt{\frac{r}{r'}}$$

This introduces a new condition. In order to enable a gain in imaging speed, $f > 1$, sHAPR must be constrained to a number of fibers less than:

$$N_f < \frac{W^2}{p^4 r'}$$

Where $r' \cdot N_f \geq 1$. For example, for our current implementation where $p = 4$ and $W = 2048$, an isotropic ($r' = 1$) imaging FOV must be smaller than 128×128 fibers to be advantageous compared to standard ROIC.

4. In the discussion on system integration, they suggest that “sHAPR integration would be especially advantageous for ... common super-resolution techniques such as single-molecule localization or structured illumination microscopy”. I can tell you from experience that SIM is *incredibly* sensitive to anything that might modify the spatial arrangement of signal in the image (such as the fiber bundle). I would be extremely surprised if this device didn't make sim reconstruction nearly impossible. And localization microscopy is so sensitive to both issues of sampling and signal collection that the bump in maximum frame rate would likely be unusable due to more other more practical limitations.

In short, I feel like the statements around the broad applicability of this technique are highly speculative, and not grounded in any particular theoretical or practical demonstration.

Response: We apologize for any confusion and concerns raised due to our statement. We fully understand the challenges and would like to clarify that our intention was mainly to highlight the core advantages of our approach in terms of speed and system integration. In our revised manuscript, we acknowledge that sHAPR currently serves as a proof-of-concept device, and its combination with various imaging strategies and platforms will undoubtedly present substantial efforts. The concerns have been partly addressed in **Supplementary Section 2.3**.

In particular, as rightly pointed out by the Reviewer, sensitive super-resolution approaches will require additional developments to address limitations related to sampling and signal. Nevertheless, it is worth noting that the utilization of fiber bundles in super-resolution microscopy is not a new concept. Fiber bundles have previously been employed to simplify the design of super-resolution systems in structured illumination microscopy⁴, super-resolution localization microscopy⁵, and image scanning microscopy⁶. For instance, the commercial Airy-scan PMT detector for Zeiss confocal super-resolution systems incorporates a fiber bundle as a relay between the objective image plane and a PMT array⁷.

In response to the Reviewer's concerns, we have refined the scope of the text, provided additional references to substantiate our claims, and, in consideration of the Reviewer's feedback on neutrality, we have further emphasized the trade-offs that potential users should consider.

Main text, lines 262-264 | Deletion of "Involving minimal alignment, we envision that a diverse range of researchers even without optical expertise can easily implement sHAPR to achieve high-speed imaging beyond 25kHz on various custom platforms, providing rich opportunities to study high-speed phenomena in IFC, cellular and molecular signaling, and other applications."

Supplementary text, lines 157-164 | Thereby, sHAPR is compatible with both label-free and fluorescence imaging, including bright-field imaging, digital holography, total internal reflection microscopy, spinning disk confocal, selective plane illumination microscopy, tomographic phase microscopy, and Fourier ptychography. Interestingly, given the recent successful application of fiber relays to techniques such as single-molecule localization⁵, structured illumination microscopy⁴, and especially image scanning microscopy^{6,7}, we anticipate the possibility of sHAPR integration with these super-resolution techniques as well.

However, users must consider key tradeoffs such as fiber losses, binning, and spatial alignments to evaluate if sHAPR will be suitable for their particular microscopy application. Several essential parameters to consider for successful integration are discussed below.

None of this is not to imply that the core concept – the re-assignment of spatial information into a format more compatible with the row-wise readout of current sCMOS cameras – is not a clever and reasonable idea. The authors have demonstrated this proof of principle and that has value. But presenting it with a bit more explicit recognition of compromises would benefit the paper, rather than make it less exciting.

These may be editorial concerns, so should the paper be accepted for publication, perhaps the editorial team could take a closer look at the neutrality of the presentation.

Response: We appreciate the comments and expect our above revisions to have facilitated the neutrality of the presentation. Furthermore, following the Reviewer's insight, we have made additional language edits in the Discussion section and addressed the inflexibility of the fiber bundle. Our goal is to provide a better guide and adoption for readers to evaluate the strengths and weaknesses of sHAPR.

Main text, lines 255-258 | Deletion of "Within the ecosystem of high-speed imaging approaches, sHAPR distinguishes itself by facilitating continuous high-speed, distortion-free, and isotropic wide-field imaging at the microsecond scale using CMOS cameras through straightforward means (**Supplementary Table S2**)."

Main text lines 254-256 | Thus, sHAPR presents a novel opportunity for users to enhance imaging speed while having the flexibility to rearrange pixels to achieve the most suitable FOV for their application.

Main text lines 261-263 | Although the bundle itself cannot be reconfigured after fabrication, sHAPR is highly versatile because it eliminates the need for mechanical components, pulsed lasers, and other hardware elements that could hinder compatibility with common imaging techniques, introduce distortions, or even limit the maximum imaging speed available.

Response to Reviewer #2 on the authors' responses to **Reviewer #3** in the previous round of reviews:

Response: First of all, we would like to extend our appreciation to Reviewer 2 for the extra effort and time spent reviewing our responses to Reviewer #3.

Regarding point 1:

> *The authors should provide more insight into the challenges of implementing other existing technologies to solve the problems of the particular field of the article.*

First, a comment about my ability to evaluate this. I am not well versed in the other fields mentioned here. It would not surprise me if there are devices in other fields that bear similarities to this one, by using fiber optics to relay an image of sorts. I think the key question here is whether anyone has mapped a 2D array of fiber optics onto a fast 1D detector (of any sort), and that should be easy enough to determine (but I personally don't know of it being done in other fields). If not, I think it's novel enough, if so, that does indeed pose a problem.

> *At least to mention how this system provides a different approach compared to similar strategies in other fields.*

I agree in a that this would be nice, but it strikes me as bit more as commentary, perhaps even something in a review or a perspective piece.

In general, I take the authors at their word that usage of fibers in other fields, while perhaps common, hasn't quite touched on the application presented here. And if that's true, I think their response is sufficient. The only bit I would take a little issue with is their vision of ease of adoption:

"we envision that a diverse range of researchers, even without optical expertise, can easily implement sHAPR to achieve high-speed imaging beyond 25kHz on various custom platforms"

I'm not sure I share that vision. I think this is a clever, but relatively niche application that may find adoption among those at the bleeding edge of speed needs (those who specifically must have 20KHz+ speeds). The rest will likely wait for camera technology itself to reach higher speeds, before reaching for an intermediate image relay. But that is a digression.

In summary, I am satisfied with the authors response to this specific point – provided that it is indeed true that other fields don't commonly map a 2D fiber array onto a 1D line detector to take advantage of speed. And that is a provision I can't personally verify.

Response: We thank the Reviewer for thoroughly examining our response. We have provided additional references on the usage of fiber optics devices to give readers a better context.

Main text, lines 253-254 | Despite the diverse applications of fiber optic relays in imaging and sensing, including endoscopy and spectroscopy, among others^{4-6,8-18}, they have not been previously exploited to exploit this feature.

Regarding point 2:

The reviewer asked:

> *The rearrangement of the camera on a given portion of the CMOS sensor for faster readout using the Fiber, is similar to changing the sensor design to read fast in a subarray of 20x40 matrix. What is the difference between doing ROIC compared to this approach? is it because of the extra time to access different rows during the sensor readout?*

This question is a bit strangely worded, but it seems to be simply asking for an explanation of why readout of an 8 x 100 pixel ROI would be faster than that of a 20 x 40 pixel ROI on a modern sCMOS. The answer is indeed as the authors say: on a CMOS camera, readout speed is limited only by the number of rows: the number of columns don't matter, and the total number of pixels doesn't matter (inasmuch as you don't change the number of rows involved in the readout). Their response to the reviewer is worded quite well I think. I also think that this point is fundamental (and commonly understood) enough that it doesn't require much additional detail in the text. The couple lines they add (47-51) are definitely sufficient.

I have one minor comment though: which is that I think the term "channels" (used by the reviewer) instead of "pixels" could be a bit confusing to many. I would use "pixels" and suggest slightly rephrase that sentence to remove the term "channels". Something like:

However, owing to the column-parallel analog-digital converter architecture typical of CMOS sensors^{49,50}, framerates are dependent only on the number of rows – not the total number of pixels – in the active subarray.

Response: We agree with the Reviewer's suggestion and have revised the cited passage in the main text to ensure consistency in our terminology for camera pixels.

Main text, lines 51 | However, owing to the column-parallel analog-digital converter architecture typical of CMOS sensors, framerates are dependent on the number of pixel rows rather than the total number of pixels.

Regarding point 3:

> It would be good if the author could explain the limitations of increasing the number of pixels from 800 to 1 million pixels (for example) to differentiate it from the ROIC procedure that converges to the sensor readout time when all the pixels of the matrix are used.

This question also makes me wonder whether the reviewer understood the basic mechanism of the speedup, and the limitations of CMOS cameras. I think the authors response is exactly right, and I commend them for pointing out that sHAPR actually gets less useful once the total number of pixels in the ROI can no longer be mapped onto a very thin (<8 row) strip of the camera. The new figure shows it nicely. This is a basic point and they address it well.

Response: We appreciate the Reviewer's confirmation.

Regarding point 4:

> does the fiber provide an extra advantage for collecting the light? compared to the sensor quantum efficiency? does it allow for optimized antireflecting coatings? curved surface? compared to having the system exposed to the surface of the sensor?

I had a related question myself, but it was in the opposite direction. I assumed the addition of the fiber would hinder light collection. On still needs to project the image onto the camera, so we haven't circumvented the quantum efficiency or coatings of the sensor here. This can likely do nothing but decrease light throughput.

The ensuing discussion about back illuminated detectors, anti-reflection coatings and microlenses on the camera, intensified cameras, curved image planes and aberration reduction, all strike me as pretty much orthogonal to the important points at hand here.

The important point here (which the authors did address to some degree as well in response to one of my points) is that the fiber bundle reduces the light throughput roughly 55%. That's a big deal when trying to go fast (it's a 2-fold penalty in speed assuming signal-to-noise ratio is held constant). If I recall correctly, the main text and/or supplement do now mention this, but I still think the authors downplay the detrimental impact of the addition of a lossy optical element into a system where speed is critical.

In any case, with respect to the reviewer here. They've answered it just fine. The answer is no, not at all :)

Response: We thank the Reviewer for the valuable feedback and commitment to ensuring that our manuscript meets the highest standards of clarity and accuracy.

References

- 1 Diebold, E. D., Buckley, B. W., Gossett, D. R. & Jalali, B. Digitally synthesized beat frequency multiplexing for sub-millisecond fluorescence microscopy. *Nature Photonics* **7**, 806-810 (2013). <https://doi.org/10.1038/nphoton.2013.245>
- 2 Hamamatsu. *Photomultiplier Tubes Basics and Applications*. 4 edn, (Hamamatsu Photonics K.K., 2017).
- 3 Wu, J. *et al.* KiloHertz two-photon fluorescence microscopy imaging of neural activity in vivo. *Nature methods* **17**, 287-290 (2020).
- 4 Bozinovic, N., Ventalon, C., Ford, T. & Mertz, J. Fluorescence endomicroscopy with structured illumination. *Optics express* **16**, 8016-8025 (2008).
- 5 Israel, Y., Tenne, R., Oron, D. & Silberberg, Y. Quantum correlation enhanced super-resolution localization microscopy enabled by a fibre bundle camera. *Nature communications* **8**, 14786 (2017).
- 6 Tenne, R. *et al.* Super-resolution enhancement by quantum image scanning microscopy. *Nature Photonics* **13**, 116-122 (2019).
- 7 Huff, J., Bathe, W., Netz, R., Anhut, T. & Weisshart, K. Technology Note: The Airyscan Detector from ZEISS Confocal Imaging with Improved Signal-to-Noise Ratio and Superresolution. (2015).
- 8 Dancker, T. A. *et al.* Multicore-fiber microendoscopy for functional cellular in-organ imaging. *bioRxiv*, 2024.2003.2002.583077 (2024). <https://doi.org/10.1101/2024.03.02.583077>
- 9 Flusberg, B. A. *et al.* Fiber-optic fluorescence imaging. *Nature methods* **2**, 941-950 (2005).
- 10 Alkhazragi, O. *et al.* Wide-field-of-view optical detectors using fused fiber-optic tapers. *Optics Letters* **46**, 1916-1919 (2021).
- 11 Ford, J. *et al.* in *Imaging Systems and Applications*. ITh1A. 4 (Optica Publishing Group).
- 12 Allington-Smith, J. Basic principles of integral field spectroscopy. *New Astronomy Reviews* **50**, 244-251 (2006).
- 13 Wang, Y., Pawlowski, M. E. & Tkaczyk, T. S. in *Imaging, Manipulation, and Analysis of Biomolecules, Cells, and Tissues IX*. 209-213 (SPIE).
- 14 Calcines, A., Harris, R. J., Haynes, R. & Haynes, D. in *Advances in Optical and Mechanical Technologies for Telescopes and Instrumentation III*. 699-718 (SPIE).
- 15 Tsikouras, A. *et al.* Streak camera crosstalk reduction using a multiple delay optical fiber bundle. *Optics letters* **37**, 250-252 (2012).
- 16 Szabo, V., Ventalon, C., De Sars, V., Bradley, J. & Emiliani, V. Spatially selective holographic photoactivation and functional fluorescence imaging in freely behaving mice with a fiberscope. *Neuron* **84**, 1157-1169 (2014).
- 17 Gilbert, J. A. The uses of fiber optics to enhance and extend the capabilities of holographic interferometry. *Holography* **10308**, 151-164 (1990).
- 18 Epstein, J. R. & Walt, D. R. Fluorescence-based fibre optic arrays: a universal platform for sensing. *Chemical Society Reviews* **32**, 203-214 (2003).

REVIEWERS' COMMENTS

Reviewer #2 (Remarks to the Author):

The revisions are great, and I thank the authors for their efforts. I think this is suitable for publication.